# Effects of *Bactericera cockerelli* Herbivory on Volatile Emissions of Three Varieties of *Solanum lycopersicum*

**DOI:** 10.3390/plants8110509

**Published:** 2019-11-15

**Authors:** Juan Mayo-Hernández, Enrique Ramírez-Chávez, Jorge Molina-Torres, María de Lourdes Guillén-Cisneros, Raúl Rodríguez-Herrera, Francisco Hernández-Castillo, Alberto Flores-Olivas, José Humberto Valenzuela-Soto

**Affiliations:** 1Departamento de Parasitología, Universidad Autónoma Agraria Antonio Narro, Buenavista, Saltillo, C.P. 25315, Coahuila, Mexico; juan_013189@hotmail.com (J.M.-H.); fdanielhc@hotmail.com (F.H.-C.); 2CINVESTAV-Unidad Irapuato, Unidad de Biotecnología e Ingeniería Genética de Plantas, Km 9.6 del Libramiento Norte Carretera Irapuato-León, Irapuato, C.P. 36821, Guanajuato, Mexico; enrique.ramirez@cinvestav.mx (E.R.-C.); jmolinat@cinvestav.mx (J.M.-T.); 3Centro de Investigación en Química Aplicada, Laboratorio Central de Instrumentación Analítica, Boulevard Enrique Reyna Hermosillo 140, Saltillo, C.P. 25294, Coahuila, Mexico; lourdes.guillen@ciqa.edu.mx; 4Universidad Autónoma de Coahuila, Facultad de Ciencias Químicas, Boulevard Venustiano Carranza y José Cárdenas S/N, Saltillo, C.P. 25280, Coahuila, Mexico; raul.rodriguez@uadec.edu.mx; 5CONACyT-Centro de Investigación en Química Aplicada, Departamento de Biocienciasy Agrotecnología, Boulevard Enrique Reyna Hermosillo 140, Saltillo, C.P. 25294, Coahuila, Mexico

**Keywords:** *Bactericera cockerelli* Sulc, headspace, volatile organic compounds, domesticated tomato plants

## Abstract

Domesticated tomato (*Solanum lycopersicum* L.) crops have presented an increased susceptibility to pests under field and greenhouse conditions. Among these pests is tomato/potato psyllid, *Bactericera cockerelli* Sulc (Hemiptera: Triozidae), a major pest in solanaceous crops. In this study, we evaluated volatile organic compound (VOC) emissions from the headspace in three healthy varieties of tomato plants (Floradade, Micro-Tom and wild) under greenhouse conditions using solid-phase microextraction and gas chromatography–mass spectrometry (SPME/GC-MS). Later, independent bioassays were performed to evaluate VOC emissions with three varieties infested with nymphs of *B. cockerelli*. The results in healthy plants showed markedly different VOC profiles in each variety (14 compounds for wild, 17 for Floradade and 4 for Micro-Tom). Plants infested with nymphs showed changes in VOC emissions distinctly in Floradade and wild varieties. We suggest that these qualitative differences in VOC profiles by the degree of domestication could explain the preferences of *B. cockerelli*.

## 1. Introduction

Plant crops can interact with many organisms in the environment, including phytopathogens and insects. Plant defenses involve different mechanisms to avoid this type of attack, and among the direct defenses, the salicylic acid (SA) pathway is activated during plant–bacteria hemibiotrophic and phloem–feeding insect interactions [1,2]. Meanwhile, the jasmonic acid/ethylene (JA/ET) pathway is activated by attacks from necrotrophic pathogens and chewing insects [3,4]. In addition, plant defenses also involve indirect mechanisms to counteract the damage caused by pathogens and herbivores [5], and these are mediated by volatile organic compounds (VOCs) with different functions in the plant.

Occasionally, the plant defenses are not sufficient in domesticated crops, and this represents a disadvantage because of their reduced capacity to respond efficiently to biotic stress in comparison with wild relative plants [6]. Wild tomato relatives produce abundant VOC-related defense compared to domesticated varieties [7]. Other studies demonstrated that the attraction of predators was reduced during the domestication process as compared with wild relatives [8]. The VOCs released by wild tomatoes are less attractive and can repel the pest (tobacco whitefly), as compared to domesticated tomatoes [9]. Although tomato plants originated in the Andean region of South America, the original site of domestication is undefined [10]. However, it is assumed that Mexico is the domestication site and Peru is the diversity center [11]. Tomato (*Solanum lycopersicum*) is an economically important crop and it is a model system for research. Among tomato models, a miniature dwarf cultivar Micro-Tom has been extensively studied for the last 10 years. Micro-Tom has a small plant size, a short life cycle and a genome size of 950 Mb, and it represents an interesting genomic tool [12,13]. On the other hand, wild tomato relative (*Solanum pennelli*) contains a larger genome size (1200 Mb) compared to commercial tomato cultivars (approximately 950 Mb). This represents a dramatic genetic erosion in domesticated cultivars [14]. Therefore, the domesticated tomatoes are susceptible to phytopathogens and phytophagous insects, and resistance alleles are present in wild tomato relatives [15].

Under field and greenhouse conditions, tomato crops (*Solanum lycopersicum*) are often exposed to phytophagous insects that impair the crop development and reduce yields. The tomato psyllid, *Bactericera cockerelli* (Hemiptera: Triozidae), is considered an important pest in the southeast USA and northeast Mexico and it attacks solanaceous crops, mainly tomato and potato (*Solanum tuberosum* L.) [16,17,18]. In North America and New Zealand, *B. cockerelli* is able to transmit the phloem-obligated proteobacteria *Candidatus* Liberibacter solanacearum (CaLso) [19]. *B. cockerelli*-CaLso in potato crops caused losses of more than $25 million between 2003 and 2005 in Texas [19,20]. However, regarding tomato defenses, CaLso induced early responses (1 week) mediated by SA, JA, and ET signaling pathways, while late responses (2 and 4 weeks) downregulated photosynthesis-associated genes, and genes involved in carbohydrate metabolism were upregulated [21]. In agreement with Casteel et al. [22], the inoculation of CaLso in the absence of its vector host suppressed the accumulation of defense transcripts (PR1 and AOS mRNAs). These results suggest that the pathogens manipulate plant defenses to benefit themselves.

To explore the feeding behavior of *B. cockerelli* on tomato plants, many factors are required for their host plant choice [23]. Tomato plants respond differently to herbivore damage by *B. cockerelli*. VOC emissions between healthy plants, and the feeding of *B. cockerelli* nymphs exhibit several changes in the blends emitted by the plants evaluated [24]. These emissions from healthy tomato plants also constitute an important signal for psyllids preference. For example, in free-choice assays performed with *B. cockerelli* adults in three varieties of tomato, domesticated plants markedly increased *B. cockerelli* oviposition, and a minor degree occurred in Micro-Tom and wild varieties [25]. These three varieties showed differences in size and branching during growth in pots (Figure 1). 

In this study, we evaluated the VOC profiles emitted by three varieties of tomato plants (*Solanum lycopersicum* cv. Floradade, *S. lycopersicum* cv. Micro-Tom and *S. lycopersicum* cv. wild), before and after *B. cockerelli* nymph feeding. Changes in VOC emissions were detected by the HS-SPME-GCMS system during tomato–*B. cockerelli* interactions. Here we found some compounds that might be involved in the *B. cockerelli* performance on tomato and this could be attributed to either the repellence or attraction of psyllids in field crops.

## 2. Results

### 2.1. Volatile Organic Compounds Profile in Healthy Plants

The profiles of VOCs collected from headspace in three different varieties of tomatoes (healthy plants) were qualitatively detected and it was possible to identify 14 VOCs for wild variety, 17 VOCs in Floradade, and 4 VOCs presented in Micro-Tom (Table 1). Among the Floradade and wild varieties, 11 compounds were shared in the VOCs blends: **1, 6, 8, 10, 13, 14, 16, 17, 18, and 19** (Table 1). Micro-Tom presented a low emission of VOCs in healthy plants; only four compounds were detected: **1, 4, 5, and 6** (Table 1).

In Floradade, only six compounds were detected as abundant (peak area > 10^8^) in healthy plants: o-cymene, (+)-4-carene, α-thujene, (+)-2-carene, ɣ-terpinene and caryophyllene. Wild relatives displayed three compounds as abundant (peak area > 10^8^): (+)-4-carene, (+)-2-carene, and β-phellandrene. In Micro-Tom, only two compounds were detected as abundant (peak area > 10^7^): α-pinene and l-β-pinene (lesser than in Floradade and wild) (Table 1).

To compare the compounds shared in the three varieties of tomato (healthy), Figure 2 shows the 15 VOCs and their changes in level among varieties. For wild and Floradade plants, compounds **6** and **10** were the major compounds emitted in psyllid-free plants although no significant differences were detected in both compounds (Figure 2, Table 2). However, compounds **18** and **20** were statistically different in emissions between wild and Floradade plants (Table 2).

### 2.2. Volatile Organic Compounds Released in Infested Plants with Bactericera cockerelli

When the three varieties of tomato were infested with nymphs of *B. cockerelli*, changes in VOC profiles emitted among varieties were evident. In Floradade, the infested leaves released a blend of volatiles which included the following: 2, 6, 7, 8, 10, 11, 14, 18, 20, and 22 (Table 1, Figure 3). For the wild type, the following volatiles were detected in the infested leaves: 1, 3, 6, 8, 9, 10, 11, 12, 14, 18, and 20 (Table 1, Figure 3). Micro-Tom showed three volatiles emitted in response to nymph feeding: 1, 5 and 18 (Table 1, Figure 3).

In Floradade–*B. cockerelli* interaction, compounds (+)-4-carene, α-thujene and (+)-sabinene (peak area > 10^8^) were the most abundant VOC emitted in response to nymph feeding (Table 1). Meanwhile, in wild–*B. cockerelli* interaction, compounds (+)-4-carene and β-phellandrene (peak area > 10^8^) were the most abundant emissions (Table 1, Figure 3). Lastly, the compounds α-pinene and β-pinene (peak area > 10^7^) were the most abundant in Micro-Tom infested after 2 h of herbivory on tomato leaves (Table 1, Figure 3).

Those plants that were infested with psyllids showed a differential profile for the eight VOCs identified, in which the compounds (+)-4-carene, β-phellandrene and (+)-sabinene (peak area > 10^8^) were the most abundant (Figure 3). Nevertheless, only the compound α-pinene was statistically different between wild and Micro-Tom plants (*P* = 0.000154).

The three varieties of tomato plants emitted qualitative and quantitative changes in VOC emissions. For example, Floradade plants presented eight compounds and the majority of these compounds were reduced in infested plants (Table 1, Figure 4B); further, seven VOCs were exclusive to healthy plants (Table 1). In wild plants, five compounds were exclusive to healthy plants and two compounds were found as exclusive to infested plants (Table 1). Micro-Tom plants presented very poor emissions in all assays performed here (Table 1).

## 3. Discussion

In this study, we evaluated VOC emissions of three varieties of tomato in healthy and psyllid-infested plants. Previous studies in our research group found that *B. cockerelli* adults presented a strong oviposition preference toward the domesticated *S. lycopersicum* cv. Floradade and a lower preference for *S. lycopersicum* cv. Micro-Tom and *S. lycopersicum* cv. wild [25]. These differences in the preferences of psyllids could be attributed to VOC blends in healthy plants. For this purpose, we performed some bioassays and the VOC profiles were obtained for healthy and infested plants (Table 1). Considering the qualitative differences among varieties of tomato VOC, we speculated that some volatiles could act as attractants or repellents for *B. cockerelli* adults, even though in this study we did not perform an electroantennography analysis to validate the results. In similar studies, healthy tomato plants (*S. lycopersicum* cv. Castlemart) emitted 11 VOCs in the constitutive blend. α-pinene, o-cymene, (+)-4-carene, and β-phellandrene were found among these compounds [24]. The shared compounds with VOC patterns from the headspace of healthy tomato plants were obtained from three varieties, although our results triggered 14 VOCs for wild, 17 VOCs for Floradade and 4 VOCs for Micro-Tom (Table 1). The oviposition behavior by *B. cockerelli* in solanaceous crops is mediated through the olfactory and visual stimuli to locate their hosts [26]. Some factors, such as natal host plant and plant phenology, also improve the settling and oviposition [23]. In other studies, wild varieties of tomato (*Solanum peruvianum*) that bore the *Mi-1.2* gene, presented a reduced oviposition of *B. cockerelli*, compared with susceptible varieties that lacked the gene [27]. In field conditions, *B. cockerelli* showed a strong preference for potato and tomato compared with eggplant and pepper [26]. 

To determine whether nymph feeding promoted differences in VOC emissions in the three varieties of tomato, two herbivory bioassays were performed on tomato leaves. In Floradade–nymph interactions, α-thujene was detected as the most abundant compound after 2 h of herbivory (Table 1). This compound was detected in basil plants emissions (*Ocimum gratissimum* L. and *O. basilicum* L.) and may be involved in the oviposition deterrence against *Tuta absoluta* [28]. Regarding VOCs induced in wild plants, β-phellandrene and (+)-4-carene were detected as the most abundant ones (Table 1). This could correlate with the reduced oviposition of *B. cockerelli* [25] and also with the role of monoterpenes involved with repellence in solanaceous plants against whiteflies, such as terpinenes and phellandrenes [9]. Interestingly, Micro-Tom plants infested with nymphs presented a reduced number of VOCs emitted, and α-pinene was the most abundant one (Table 1). Besides, we did not perform leaves surface analysis to evaluate the differences in trichome numbers. The synthesis of diverse trichome-borne metabolites influences host plant selection by herbivores under natural conditions [29]. Prager et al. [30] showed that total alkaloid and phenolics levels were greater in three breeding lines of potato compared with commercial cultivar. These differences might explain the reduced psyllid oviposition and aphid reproduction observed. Although our experiments were performed with *B. cockerelli* CaLso-positive, 2 h of herbivory by nymphs (4th and 5th stages) was sufficient to induce VOC emissions for leaf damage. In this sense, to transmit CaLso in solanaceous, it is necessary to have at least 24 h of feeding by *B. cockerelli* [31]. CaLso manipulates host tomato defenses within 1 week, and when the pathogen population is increased in the phloem, it can evoke changes in the expression of defense genes [21]; this modification of gene expression may induce changes in VOC emissions to attract the psyllid vector [32]. Therefore, we speculate that the changes in VOC emissions in each tomato variety were promoted by *B. cockerelli* herbivory and not by CaLso. Mustafa et al. [33] showed that bittersweet nightshade (*Solanum dulcamara*) improved the development and body size of three haplotypes of *B. cockerelli* (Hemiptera: Triozidae) compared to potato plants. Lately, the study of wild relatives has been expanded to understand those mechanisms involved for indirect defense against phytophagous insects, with an aim to improve the commercial crops.

## 4. Materials and Methods

### 4.1. Plant Growth

Tomato seeds (*Solanum lycopersicum*) from three varieties were germinated in peat-moss until two leaves were presented. Seedlings from domesticated (*S. lycopersicum* cv. Floradade, Crown Seeds, Homestead, FL, USA), Micro-Tom (*S. lycopersicum* cv. Micro-Tom), and wild type (*S. lycopersicum* cv. wild) were transplanted in 1.5 L pots containing peat-moss:perlite (70:30 v/v). The plants were watered every 3 days and fertilized once a week with Steiner’s nutritive solution (25%). Tomato plants were maintained under greenhouse conditions at 28 ± 2 °C, RH = 45%, and were allowed to grow until they reached 20 cm height or had four to five fully extended leaves for the bioassays. The seeds of Micro-Tom employed were kindly provided by Dr. John Délano-Frier (CINVESTAV-Unidad Irapuato), whereas the wild type seeds were collected in summer 2015 in Irapuato (20°34′34.3″ N 101°23′24.3″ W), Guanajuato, Mexico. Wild type tomato was taxonomically characterized as a member of *Solanum lycopersicum* and was deposited with catalogue number XAL0148356 in Herbario XAL by Carlos Durán Espinoza from Instituto de Ecología A.C. in Veracruz, Mexico.

### 4.2. Insect Rearing

The potato/tomato psyllids (*Bactericera cockerelli* Sulc) were reared on tomato plants under greenhouse conditions. *B. cockerelli* (CaLso positive) was maintained in cages of 50 × 50 × 50 cm with the same conditions described above. Psyllids were donated by Dr. Alberto Flores from Departamento de Parasitología of Universidad Autónoma Agraria Antonio Narro (UAAAN), in Saltillo, Mexico.

### 4.3. Headspace Collection and Analysis of Plant Volatiles

Three varieties of tomato plants (2 weeks after transplanting) were selected to collect the volatile emissions (healthy plants). Firstly, to obtain volatile profiles, we employed the headspace–solid-phase microextraction and gas chromatography–mass spectrometry (HS-SPME-GCMS) system. Individual plants were placed into a polypropylene desiccator previously adapted with a charcoal-activated trap, and flow-air (6 L/min) was passed on the plant for 5 min (Floradade); polyethylene bags (Sunbag, MERCK KGaA, Darmstadt, Germany) were utilized for Micro-Tom and wild. To obtain the VOC profiles, three varieties with three biological replicates (*n* = 3) were performed for a single bioassay in healthy tomato plants (psyllid-free).

To evaluate the VOCs released in the three varieties during *B. cockerelli* nymph feeding, two bioassays were performed independently with three biological replicates for all varieties. With a small brush, 25 nymphs of *B. cockerelli* (fourth and fifth stages) were gently placed on each plant and allowed 2 h feeding on the tomato leaves (nymphs were left in plants). After feeding, individual plants were collected, using either bags or a desiccator, to evaluate the headspace of each treatment by SPME. For SPME, gray fiber (50/30 µm, DVB/CAR/PDMS, MERCK KGaA, Darmstadt, Germany) was used in all evaluations. The fiber was exposed during 2.5 h. Healthy plants (nymphs-free) were exposed at the same time for SMPE. The bioassays were performed at 28 ± 2 °C in greenhouse conditions.

All fibers were injected and desorbed in a gas chromatograph (GC 7890A, Agilent Technologies, Wilmington, DE, USA) coupled to mass selective detector (Agilent Technologies MSD 5975 C). All analyses were performed under running conditions: initial oven temperature of 60 °C for 1 min, increasing 8 °C/min until 280 °C for 1 min. Injector temperature was 230 °C. The VOCs were analyzed employing a capillary column Agilent Technologies HP-5MS (30 m × 0.25 mm × 0.25 µm). After 15 minutes, the fiber was removed from the injector and stored until use. All the compounds were identified by a NIST92 and Wiley library, according the percentage of quality presented (> 90%); retentions times were compared with library, and Kovats indices also were determined for individual compounds.

### 4.4. Data Analysis

All VOCs collected from headspace of the three varieties of tomato (healthy and infested) were graphed using GraphPad Prism version 5 for Windows, GraphPad Software, La Jolla, California USA. Prior to analysis, the raw data of peak areas were log transformed and were tested using univariate ANOVA, the non-parametric *Kruskal-Wallis* test. Statistical analyses were performed using R Statistical Software (R Core Team 2004).

## Figures and Tables

**Figure 1 plants-08-00509-f001:**
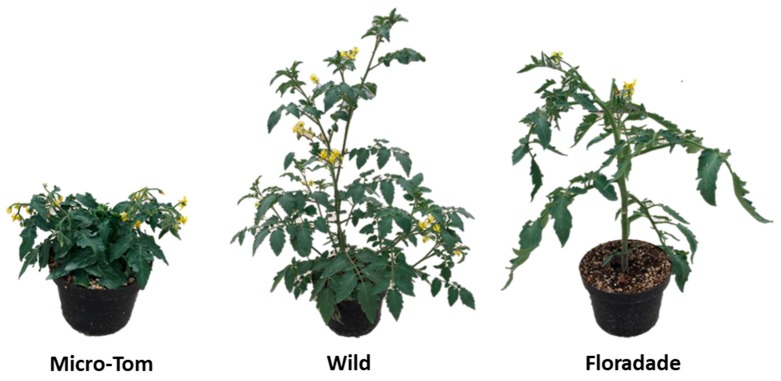
Differences presented between three varieties of tomato (*Solanum lycopersicum*). Plants were grown in pots during 4 weeks under greenhouse conditions.

**Figure 2 plants-08-00509-f002:**
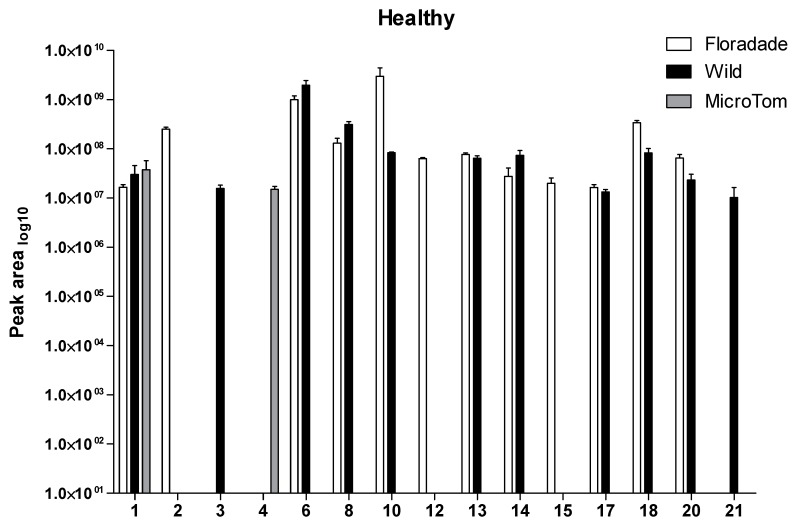
Plant volatiles from healthy plants of three varieties of tomato. Compounds were presented in profile and were obtained from three varieties for one bioassay in healthy tomato plants (psyllid-free). Floradade (white bars), wild (black bars), and Micro-Tom (gray bars). Compounds with > 90% of quality were selected: α-pinene (**1**), o-cymene (**2**), 1,3,5-Cycloheptatriene, 3,7,7-trimethyl- (**3**), l-β-pinene (**4**), (+)-4-carene (**6**), (+)-2-carene (**8**), ɣ-terpinene (**10**), α-terpinolene (**12**), ascaridole (**13**), δ-elemene (**14**), copaene (**15**), isocaryophyllene (**17**), caryophyllene (**18**), humulene (**20**), and hexadecane (**21**). Values are means ± SE, *n* = 3.

**Figure 3 plants-08-00509-f003:**
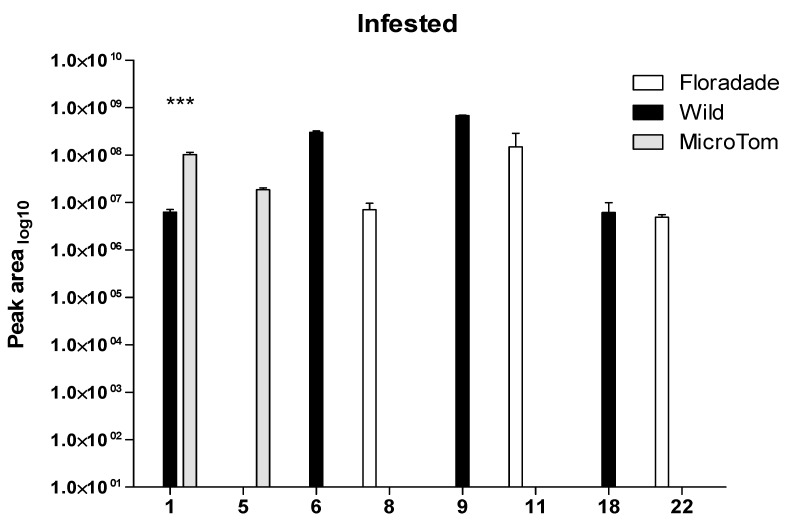
Plant volatiles from infested plants of three varieties of tomato. Compounds were presented in profile and were obtained from the three varieties for two bioassays in infested tomato plants: Floradade (white bars), wild (black bars), and Micro-Tom (gray bars). Compounds: α-pinene (**1**), β-pinene (**5**), (+)-4-carene (**6**), (+)-2-carene (**8**), β-phellandrene (**9**), (+)-sabinene (**11**), caryophyllene (**18**), and phenol, 2,6-bis(1,1-dimethylethyl)-4-(1-ethylpropyl)- (**22**). Values are means ± SE, *n* = 6. Level of significance: ***, *P* < 0.001. (Univariate ANOVA, non-parametric *Kruskal-Wallis* test at *P* = < 0.05).

**Figure 4 plants-08-00509-f004:**
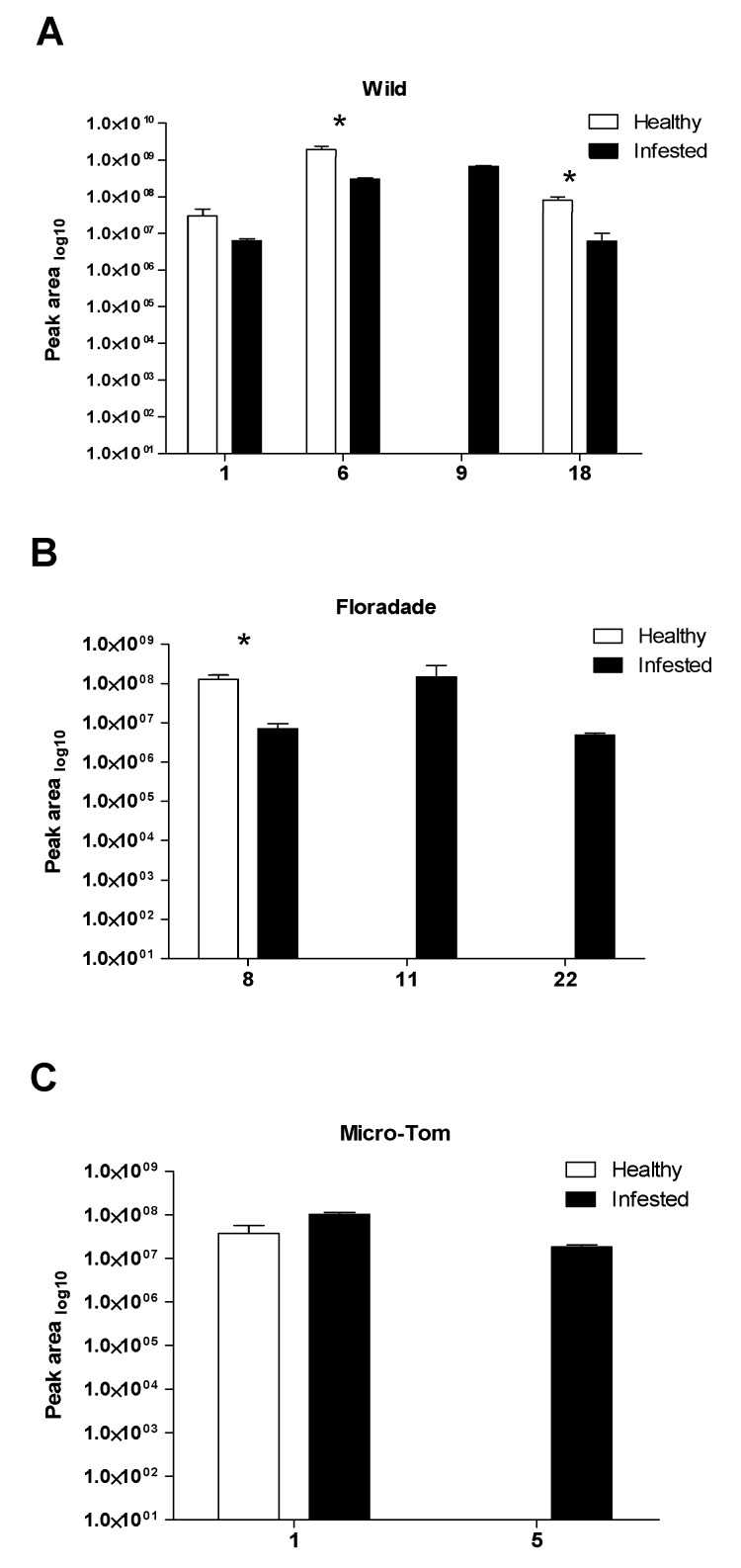
Volatile organic compounds detected in *Solanum lycopersicum* varieties (wild, Floradade and Micro-Tom) from healthy (white bars) and infested (black bars) plants. (**A**) Wild plants with compounds: α-pinene (1), (+)-4-carene (6), β-phellandrene (9), and caryophyllene (18). (**B**) Floradade plants with compounds: (+)-2-carene (8), (+)-sabinene (11), and phenol, 2,6-bis(1,1-dimethylethyl)-4-(1-methylpropyl) (22). (**C**)Micro-Tom plants with compounds: α-pinene (1), and β-pinene (5). VOC profiles were obtained from the three varieties for a single bioassay in healthy plants (psyllid-free) and two bioassays in infested tomato plants. Values are means (±SE; healthy *n* = 3; infested *n* = 6). Level of significance: *, *P* < 0.05. (Univariate ANOVA, non-parametric *Kruskal-Wallis* test at *P* = < 0.05).

**Table 1 plants-08-00509-t001:** List of volatile organic compounds identified from headspace of healthy and infested plants with *Bactericera cockerelli* nymphs (wild, Floradade and Micro-Tom). VOC profiles were obtained from three varieties with three biological replicates for one bioassay in healthy plants (psyllid-free) and two bioassays in infested tomato plants. Values indicate means (± SD; healthy *n* = 3; infested *n* = 6). Data with no SD were only detected in one replicate. -, not detected. Peak areas 10^6^ to 10^9^ were log10-transformed. Bold numbers indicate significant differences in healthy plants. (Univariate ANOVA, non-parametric *Kruskal-Wallis* test at *P* ≤ 0.05).

		RT	Wild		Floradade		Micro-Tom	
Compound		(min)	Healthy	Infested	Healthy	Infested	Healthy	Infested
**1**	α-Pinene	4.568	7.28 ± 0.46	6.78 ± 0.09	7.21 ± 0.09	-	7.43 ± 0.36	8.00 ± 0.07
**2**	o-Cymene	5.157	-	-	8.39 ± 0.06	6.96	-	-
**3**	1,3,5-Cycloheptatriene, 3,7,7-trimethyl-	5.232	7.18 ± 0.11	6.47	-	-	-	-
**4**	l-β-Pinene	5.275	-	-	-	-	7.17 ± 0.07	-
**5**	β-Pinene	5.344	-	-	-	-	6.53	7.26 ± 0.06
**6**	(+)-4-Carene	5.615	9.26 ± 0.18	8.48 ± 0.03	8.97 ± 0.13	7.66	6.98	-
**7**	α-Thujene	5.692	-	-	8.78	8.46	-	-
**8**	(+)-2-Carene	5.771	8.49 ± 0.06	7.25	8.10 ± 0.11	6.81 ± 0.17	-	-
**9**	β-Phellandrene	6.166	9.55	8.83 ± 0.01	-	-	-	-
**10**	ɣ-Terpinene	6.193	7.92 ± 0.02	7.63	9.02 ± 0.88	6.48	-	-
**11**	(+)-Sabinene	6.413	-	7.94	6.89	7.67 ± 0.79	-	-
**12**	α-Terpinolene	7.224	-	7.69	7.79 ± 0.03	-	-	-
**13**	Ascaridole	10.463	7.80 ± 0.05	-	7.88 ± 0.03	-	-	-
**14**	δ-Elemene	11.523	7.83 ± 0.16	7.18	7.29 ± 0.39	6.78	-	-
**15**	Copaene	12.207	-	-	7.26 ± 0.17	-	-	-
**16**	(−)-cis-β-Elemene	12.452	7.07	-	7.41	-	-	-
**17**	Isocaryophyllene	12.715	7.12 ± 0.05	-	7.21 ± 0.06	-	-	-
**18**	Caryophyllene	12.914	**7.89 ± 0.13**	6.61 ± 0.37	**8.52 ± 0.06**	7.50	-	6.63
**19**	Bicyclo[7.2.0]undecane, 10,10-dimethyl-2,6-bis(methylene)-, [1S-(1R*,9S*)]-	13.065	7.25	-	7.30	-	-	-
**20**	Humulene	13.468	**7.32 ± 0.19**	6.60	**7.79 ± 0.12**	6.85	-	-
**21**	Hexadecane	15.827	6.91 ± 0.30	-	-	-	-	-
**22**	Phenol, 2,6-bis(1,1-dimethylethyl)-4-(1-ethylpropyl)-	16.142	-	-	6.48	6.68 ± 0.06	-	-
	**Number of compounds**		14	11	17	10	4	3

**Table 2 plants-08-00509-t002:** Statistical analyses for the compounds that were shared in the headspace of healthy plants among, at least, two out of three varieties (those compounds with > 90% of quality were selected). Values indicate non detected means (± SD; *n* = 3). *P* values indicate significant differences in compounds levels among the wild, Floradade and Micro-Tom varieties. ns, non-significant. Level of significance: *, *P* = 0.05; **, *P* = 0.01. #, number of compound assigned (see Table 1). (Univariate ANOVA, non-parametric *Kruskal-Wallis* test at *P* = < 0.05).

#	Compound	Wild	Floradade	Micro-Tom	*P* Value	*F* Value	Df
**1**	α-Pinene	7.287 ± 0.46	7.21 ± 0.09	7.437 ± 0.36	0.8021 (ns)	0.2288	2
**6**	(+)-4-Carene	9.260 ± 0.18	8.979 ± 0.13	-	0.1501 (ns)	3.1594	1
**8**	(+)-2-Carene	8.490 ± 0.06	8.100 ± 0.11	-	0.0955 (ns)	8.9951	1
**10**	ɣ-Terpinene	7.922 ± 0.02	9.026 ± 0.88	-	0.1493 (ns)	3.1757	1
**13**	Ascaridole	7.807 ± 0.05	7.883 ± 0.03	-	0.3418 (ns)	1.5289	1
**14**	δ-Elemene	7.838 ± 0.16	7.296 ± 0.39	-	0.1420 (ns)	3.331	1
**17**	Isocaryophyllene	7.121 ± 0.05	7.212 ± 0.06	-	0.3764 (ns)	1.2728	1
**18**	Caryophyllene	7.893 ± 0.13	8.529 ± 0.06	-	0.0037 **	36.783	1
**20**	Humulene	7.323 ± 0.19	7.797 ± 0.12	-	0.0386 *	9.1983	1

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
