# Peer review of "Effects of Bactericera cockerelli Herbivory on Volatile Emissions of Three Varieties of Solanum lycopersicum"

_plants, 2019, doi:10.3390/plants8110509_

Round 1

Reviewer 1 Report

Review: Juan Mayo-Hernández et al.: “Effects of Volatile Emissions on Three Varieties of Solanum lycopersicum Affected by Bactericera cockerelli herbivory”; submitted to Plants

 In their manuscript submitted to Plants Mayo-Hernández et al. investigate the effect of plant domestication on both constitutive and herbivore-induced volatile organic compound (VOC) emissions from three tomato cultivars along a domestication gradient. Constitutive VOC emissions were evaluated in a first experiment by sampling the headspace of non-infested plants using solid-phase microextraction (SPME)-GC-MS. In a second experiment, individual plants were infested by 25 nymphs of the tomato psyllid, Bactericera cockerelli, which were allowed to feed for a 2-hour period prior to sampling of the plants’ headspace.

Concerning the constitutive VOC emissions the authors observed both qualitative and quantitative differences in the volatile profiles of the three varieties with the most pronounced differences being observed between the cultivar ‘Micro-Tom’, which emitted lower VOC quantities in general, and the other two cultivars, ‘Floradade’ and ‘Wild Type’, respectively. Following infestation by B. cockerelli, VOC emissions tended to be lower for most of the detected compounds in ‘Floradade’ and ‘Wild Type’ compared to non-infested plants, whereas ‘Micro-Tom’ appeared to emit higher amounts, although again at a much lower level than the other two cultivars. Based on their findings the authors conclude that the observed “qualitative differences in VOC profiles by the degree of domestication could explain the preferences of B. cockerelli.”

Regarding the quality of the manuscript I do have several major concerns including experimental procedures and data analysis/interpretation as well as the integration of the observations made by the authors into the current understanding of the tomato/B. cockerelli interaction. Therefore, I cannot recommend the publication of the manuscript in its current state but rather would encourage the authors to resubmit an extensively revised and, hopefully, improved version.

In the following I will focus on the most pressing concerns only and ignore the minor ones concerning the language used, spelling etc. Still, the manuscript also needs some considerable improvements concerning the latter.

Major comments:

Title:

The title is misleading as is now. I would recommend a rearrangement of the words: “Effects of Bactericera cockerelli herbivory on volatile emissions of three varieties of Solanum lycopersicum”; and also include the term domestication since this is the scope of the manuscript.

Study design:

I could only find scarce information on how many replicates were done for each experiment. For the first experiment please state the exact number of for each cultivar in the methods section. For the second experiment the only information I found was that two bioassays were performed. I wonder whether this equals to two replicates. In addition to mentioning this in the methods section, I would wish for this information in the figure captions and also in the tables that are presented. To me the difference between the first experiment on healthy plants and the controls of the second experiment is unclear. As I understood it, in both cases non-infested plants were sampled by SPME for 2.5 hours. This should be clarified. Also, if the “treatments” are the same, were the same compounds detected in the headspace? Because for the second experiment only those are presented that were detected in infested plants. How the cultivars were selected is also unclear to me. Here, I would wish for more information on the domestication process of tomato and which species/cultivar is considered the ancestral one; especially for the reader who is not familiar with the topic. In Bergougnoux 2014 (Biotechnology Advances), for instance, the following information can be found:

“The most likely ancestor of cultivated tomatoes is the wild cherry tomato, usually identified as S. lycopersicum var. cerasiforme because of its wide representation in Central America. Nevertheless the genetic investigations made by Nesbitt and Tanksley in 2002 demonstrated that the plants known as “cerasiforme” are a mixture of wild and cultivated tomatoes, rather than the direct ancestor of the cultivated tomato. A very recent study based on the analysis of single nucleotide polymorphisms not only confirms that S. lycopersicum var. cerasiforme is not the ancestor of the cultivated tomato but also reinforces the model that a predomestication of the tomato occurred in the Andean region (Peruvian hypothesis), with the domestication being completed in Mesoamerica (Mexican hypothesis), followed by its introduction to Europe by Spaniards and then spread all over the world (Blanca et al.,2012 ).”

In my opinion, it should be clarified how the used ‘wild-type’ relates to this. Is it also mixture/hybrid of wild and cultivated tomatoes? In addition, it should be mentioned to what extent and why ‘Floradade’ and ‘Micro-Tom’ differ in their degree of domestication. To my (maybe limited) knowledge, both are commercial cultivars and thus the product of extensive breeding.

Lastly, I was wondering why an infestation period of only 2 hours was chosen since in many publications that investigated volatile emissions upon infestation by phloem-feeders, and also following tomato psyllid feeding on tomato plants (e.g. Bautista-Lozada et al. 2013 – PLOS One), longer infestation periods of up to several days were chosen.

Material and methods:

As mentioned above, I miss information on the exact study design. How many replicates were done? What were the differences between the healthy plants of experiment 1 and experiment 2? Analytics: I miss information on how the individual compounds were identified? Which library were the mass spectra compared to? Were the retention times and mass spectra compared to an in-house library or authenticated standards? Were Kovats indices determined for the individual compounds to assist identification? Please clarify! Related to the previous point: the authors might include a table in the supporting information where the retention times, Kovats indices, if determined, and major m/z for each compound are reported; alongside with an exemplary and labelled chromatogram.

Results and data analysis:

In general, I would recommend to structure the results section using appropriate sub headings. For this manuscript this might just be two, one for constitutive emissions and one for induced emissions. Nevertheless, structure will help the reader to capture the main messages better. In the current form, the results go from constitutively released volatiles to induced emissions and back to constitutive emissions.

Please number all compounds consecutively from 1 to n across tables 1 and 2 and stick to that numbering in all figures. This will make it a lot easier for the reader to compare between figures and tables. As a recommendation: since all peak area units given for the individual compounds are arbitrary anyway, and range across app. four orders of magnitude, the authors might as well just use numbers in the range from 0.1 to 100. In my opinion, this is much more intuitive to read and will also be useful for the next point. Concerning the two tables, I would prefer to see actual numbers, e.g. mean±SD, instead of “+”, “++” etc.

Concerning the data presentation, I also have a few suggestions:

Table 2: I would recommend to design the table in the following format

RT

Wild

Floradade

Micro-Tom

Compound

(min)

healthy

infested

healthy

infested

healthy

Infested

1

α-Pinene

4.574

25±5

22±4

etc…

and give mean peak areas ± SD.

Tables 1 and 2: Please include a line in the end giving the total emissions for each column. Was it considered to perform a multivariate analysis on the observed volatile profiles of healthy plants, e.g. a principal component analysis (PCA) or a partial least squares-discriminant analysis (PLS-DA, e.g. see Fatouros et al. 2012 – PLOS One; https://journals.plos.org/plosone/article?id=10.1371/journal.pone.0043607) instead of showing the measured emissions for a selection of 11 compounds for the three cultivars (Figure 1)?

This might yield some more insight whether the profiles actually differ between the three cultivars. Assuming, of course, that a valid model can be built with the number of replicates at hand.

In my opinion, a composite figure showing the total emissions, including an appropriate statistical test, and the outcome of the PCA would then give some good idea in how far the different cultivars actually differ and whether the domestication gradient is represented in the volatile profiles.

The same applies to the second experiment. Although, as it seems to me for the moment, only two replicates have been done, which will not suffice for this kind of analysis.

Concerning the number of replicates: I wonder how t-tests were done for the data shown in figures 4 and 5? It is stated that “boxes with no bars indicate no data were collected for one assay”. And there are quite a few boxes without bars suggesting that there was only one replicate. With this low number of replicates a non-parametric test might be more appropriate, anyway. Concerning figures 3 to 5: How were the depicted compounds selected? Since only 11, 12, and 4 compounds were reported for infested ‘Wild’, ‘Floradade’ and ‘Micro-Tom’, respectively, in table 2, would it be possible to show all of them in the figures?

Theoretical background & data interpretation:

In ll. 200 it is stated: “With the qualitative differences among varieties of tomato VOC we speculated that some volatiles could act as attractants or repellents for cockerelli adults, although in this study we did not perform electroantennography analysis to validate the results.” First of all, it is not mentioned which compounds were speculated on as attractants or repellents. Is there anything known on individual compounds that are attractive or repellent to B. cockerelli? Would it be possible to test whether B. cockerelli adults display preferences for any of the cultivars used here? Could the authors construct a simple y-tube olfactometer (e.g. as in Fatouros et al. 2012 – PLOS One, Figure 5) to test this? In this kind of setup maybe even individual compounds could be tested. While these assays are not strictly required for the present manuscript, they would considerably improve it and strengthen the hypothesis that the host plant choices that have been observed earlier are indeed mediated by contrasting volatile emissions. Especially so, since the host plant preferences that are referred to were observed during cage experiments where the insects can directly access the plants and volatiles may thus only play a minor role for host plant acceptance. Overall, ‘Floradade’ and ‘Wild type’ appear very similar to me, both in their constitutive emissions but also considering the volatile emissions following psyllid infestation. As shown in Figures 1 and 2 the released amounts are very similar for the shared compounds. In addition, in both varieties, the relative decrease in γ-terpinene, caryophyllene and humulene emissions following psyllid infestation also appears very similar. By contrast, ‘Micro-Tom’ does not only emit a strikingly smaller number of compounds, but following psyllid infestation α-Pinene emissions, for instance, are as high as in healthy plants in contrast to the ‘Wild’ cultivar, for which α-Pinene emissions appear to decrease (although this decrease was not statistically significant). Therefore, I do not think that domestication explains the observed differences well enough given the similarity between ‘Floradade’ and ‘Wild’ and assuming that ‘Floradade’ and ‘Micro-Tom’ represent domesticated cultivars. Is there anything known on natural enemies on tomato psyllids when feeding on tomato plants and whether they use olfactory cues to detect their prey/hosts?

Author Response

Reviewer 1:

Major comments:

Title:  The title is misleading as is now. I would recommend a rearrangement of the words: “Effects of Bactericera cockerelli herbivory on volatile emissions of three varieties of Solanum lycopersicum”; and also include the term domestication since this is the scope of the manuscript.

We agree with this observation. The title was thereby changed to: “Effects of Bactericera cockerelli herbivory on volatile emissions of three varieties of Solanum lycopersicum”.

Study design:

1) I could only find scarce information on how many replicates were done for each experiment. For the first experiment please state the exact number of for each cultivar in the methods section. For the second experiment the only information I found was that two bioassays were performed. I wonder whether this equals to two replicates. In addition to mentioning this in the methods section, I would wish for this information in the figure captions and also in the tables that are presented.

Changes were made as suggested (L276-278 and L280). Tables and figures captions were also modified.

2) To me the difference between the first experiment on healthy plants and the controls of the second experiment is unclear. As I understood it, in both cases non-infested plants were sampled by SPME for 2.5 hours. This should be clarified. Also, if the “treatments” are the same, were the same compounds detected in the headspace? Because for the second experiment only those are presented that were detected in infested plants.

The table 1 show the VOC profiles identified of healthy plants in three varieties by SPME for 2.5 hours. VOC profile presented in table 2 were identified of infested plants in all varieties for 2.5 hours, although for figures 2, 3, 4 and 5 those compounds that were presented in infested varieties compared with the basal levels in healthy plants (first bioassay).

3) How the cultivars were selected is also unclear to me. Here, I would wish for more information on the domestication process of tomato and which species/cultivar is considered the ancestral one; especially for the reader who is not familiar with the topic. In Bergougnoux 2014 (Biotechnology Advances), for instance, the following information can be found:

“The most likely ancestor of cultivated tomatoes is the wild cherry tomato, usually identified as S. lycopersicum var. cerasiforme because of its wide representation in Central America. Nevertheless the genetic investigations made by Nesbitt and Tanksley in 2002 demonstrated that the plants known as “cerasiforme” are a mixture of wild and cultivated tomatoes, rather than the direct ancestor of the cultivated tomato. A very recent study based on the analysis of single nucleotide polymorphisms not only confirms that S. lycopersicum var. cerasiforme is not the ancestor of the cultivated tomato but also reinforces the model that a predomestication of the tomato occurred in the Andean region (Peruvian hypothesis), with the domestication being completed in Mesoamerica (Mexican hypothesis), followed by its introduction to Europe by Spaniards and then spread all over the world (Blanca et al.,2012 ).”

In my opinion, it should be clarified how the used ‘wild-type’ relates to this. Is it also mixture/hybrid of wild and cultivated tomatoes? In addition, it should be mentioned to what extent and why ‘Floradade’ and ‘Micro-Tom’ differ in their degree of domestication. To my (maybe limited) knowledge, both are commercial cultivars and thus the product of extensive breeding. 4) Lastly, I was wondering why an infestation period of only 2 hours was chosen since in many publications that investigated volatile emissions upon infestation by phloem-feeders, and also following tomato psyllid feeding on tomato plants (e.g. Bautista-Lozada et al. 2013 – PLOS One), longer infestation periods of up to several days were chosen.

The cultivars of tomato were selected for availability in our laboratory, ‘Floradade’ (Crown Seeds) was selected as a commercial cultivar commonly used in greenhouse and field conditions. ‘Micro-Tom’ was selected as a cultivar extensively studied in laboratory and greenhouse level, mainly because transgenic and mutants plants were obtained with this background genetic. ‘Wild-type’ was selected according their tolerance to herbivorous insects observed in the environment and was collected with the main objective to evaluate those traits involved in their reduced preference oviposition of Bactericera cockerelli (Mayo Hernández et al. 2018). For us, the degree of domestication in each cultivar of tomato was not determined in our laboratory we expected that ‘Floradade’ is more attractive to pests than ‘Wild-type’ tomato plants.

The bioassays were performed with nymphs of psyllid that harboring the pathogen Candidatus Liberibacter solanacearum (Calso) and 2 hours was chosen for feeding regarding to previous reports in the literature mentioned that Calso could manipulated the plant responses after 24 hours [Sengoda, V.G.; Cooper, W.R.; Swisher, K.D.; Henne, D.C.; Munyaneza, J.E. Latent period and transmission of “Candidatus Liberibacter solanacearum” by the potato psyllid Bactericera cockerelli (Hemiptera: Triozidae). PLoS One. 2014, 9:e93475.]

Material and methods:

1) As mentioned above, I miss information on the exact study design. How many replicates were done? What were the differences between the healthy plants of experiment 1 and experiment 2? 

Changes were made as suggested (L276-278 and L280).

2) Analytics: I miss information on how the individual compounds were identified? Which library were the mass spectra compared to? Were the retention times and mass spectra compared to an in-house library or authenticated standards? Were Kovats indices determined for the individual compounds to assist identification? Please clarify! 

All the compounds were identified by a NIST92 and Wiley library, according the percentage presented (>90%), retentions times were compared with library and Kovats indices also were determined for individual compounds.

3) Related to the previous point: the authors might include a table in the supporting information where the retention times, Kovats indices, if determined, and major m/z for each compound are reported; alongside with an exemplary and labelled chromatogram.

We agree with comment, but we considered that the information in tables could be enough to explain those differences between varieties of tomato.

Results and data analysis:

In general, I would recommend to structure the results section using appropriate sub headings. For this manuscript this might just be two, one for constitutive emissions and one for induced emissions. Nevertheless, structure will help the reader to capture the main messages better. In the current form, the results go from constitutively released volatiles to induced emissions and back to constitutive emissions.

Changes were made as recommended.

1) Please number all compounds consecutively from 1 to n across tables 1 and 2 and stick to that numbering in all figures. This will make it a lot easier for the reader to compare between figures and tables.

Changes were made as suggested.

2) As a recommendation: since all peak area units given for the individual compounds are arbitrary anyway, and range across app. four orders of magnitude, the authors might as well just use numbers in the range from 0.1 to 100. In my opinion, this is much more intuitive to read and will also be useful for the next point.

We selected the symbols (+, ++…etc.) in order to explain the possible differences in abundance between varieties for table 1 and 2. Also, as mentioned above some compounds were detected only once.

3) Concerning the two tables, I would prefer to see actual numbers, e.g. mean±SD, instead of “+”, “++” etc. Concerning the data presentation, I also have a few suggestions:

4) Table 2: I would recommend to design the table in the following format

 RT Wild Floradade Micro-Tom Compound (min) healthy infested healthy infested healthy Infested 1 α-Pinene 4.574 25±5 22±4 etc…    and give mean peak areas ± SD.

Please, see above comment.

5) Tables 1 and 2: Please include a line in the end giving the total emissions for each column. 

Changes were made as recommended.

6) Was it considered to perform a multivariate analysis on the observed volatile profiles of healthy plants, e.g. a principal component analysis (PCA) or a partial least squares discriminant analysis (PLS-DA, e.g. see Fatouros et al. 2012 – PLOS One; https://journals.plos.org/plosone/article?id=10.1371/journal.pone.0043607) instead of showing the measured emissions for a selection of 11 compounds for the three cultivars (Figure 1)? This might yield some more insight whether the profiles actually differ between the three cultivars. Assuming, of course, that a valid model can be built with the number of replicates at hand.  In my opinion, a composite figure showing the total emissions, including an appropriate statistical test, and the outcome of the PCA would then give some good idea in how far the different cultivars actually differ and whether the domestication gradient is represented in the volatile profiles. The same applies to the second experiment. Although, as it seems to me for the moment, only two replicates have been done, which will not suffice for this kind of analysis.

We agree with comments, in different compounds either in healthy and infested plants had not minimal replicates obtained for statistical analysis. Regarding the reference (Fatouros et al. 2012), they evaluated volatile emissions measured as peak area divided by the fresh weight of a plant’s foliage using the software SIMCA P+12.0, however, in our bioassays performed we did not weight the fresh foliar tissue.

7) Concerning the number of replicates: I wonder how t-tests were done for the data shown in figures 4 and 5? It is stated that “boxes with no bars indicate no data were collected for one assay”. And there are quite a few boxes without bars suggesting that there was only one replicate. With this low number of replicates a non-parametric test might be more appropriate, anyway.

We agree with comment. The figures were corrected and error bars were removed to avoid confusion in replicates number.

8) Concerning figures 3 to 5: How were the depicted compounds selected? Since only 11, 12, and 4 compounds were reported for infested ‘Wild’, ‘Floradade’ and ‘Micro-Tom’, respectively, in table 2, would it be possible to show all of them in the figures?

Total compounds were selected for their abundance with >90% presented after GC-MS analysis.

Theoretical background & data interpretation:

1) In ll. 200 it is stated: “With the qualitative differences among varieties of tomato VOC we speculated that some volatiles could act as attractants or repellents for B. cockerelli adults, although in this study we did not perform electroantennography analysis to validate the results.” First of all, it is not mentioned which compounds were speculated on as attractants

or repellents. Is there anything known on individual compounds that are attractive or repellent to B. cockerelli? Would it be possible to test whether B. cockerelli adults display preferences for any of the cultivars used here? Could the authors construct a simple y-tube olfactometer (e.g. as in Fatouros et al. 2012 – PLOS One, Figure 5) to test this? In this kind of setup maybe even individual compounds could be tested. While these assays are not strictly required for the present manuscript, they would considerably improve it and strengthen the hypothesis that the host plant choices that have been observed earlier are indeed mediated by contrasting volatile emissions. Especially so, since the host plant preferences that are referred to were observed during cage experiments where the insects can directly access the plants and volatiles may thus only play a minor role for host plant acceptance. 

We speculated that (+)-4-carene and β-phellandrene could act as repellent in wild plants against to B. cockerelli, although we considered in future experiments to found by Y-tube olfactometer and electroantennography analysis those compounds involved either in repellence or attraction. Regarding to repellent compounds, we found dimethyl sulfide (DMDS) and five plant essential oils (thyme, tea tree, peppermint, savory and clove), DMDS and plant oils had a significant repellent effect on adults of B. cockerelli, however, individual compounds were not determined in this study (John Diaz-Montano and John T. Trumble. 2013. Behavioral responses of the potato psyllid (Hemiptera: Triozidae) to volatiles from dimethyl sulfide and plant essential oils. J Insect Behav 26:336-351). Recently, Mayo-Hernández et al. (2018) reported that high preference of oviposition by B. cockerelli was presented in ‘Floradade’ and ‘Wild type’ showed a lower preferences (Mayo-Hernández, J.; Flores-Olivas, A.; Valenzuela-Soto, J.H.; Rodríguez-Pagaza, Y.; Vega-Chávez, J.; Hernández-Castillo, F.; Aguirre-Uribe, L. Bactericera cockerelli Sulc oviposition preference and development on three tomato varieties. Southwest Entomol. 2018, 43, 905–910).

2) Overall, ‘Floradade’ and ‘Wild type’ appear very similar to me, both in their constitutive emissions but also considering the volatile emissions following psyllid infestation. As shown in Figures 1 and 2 the released amounts are very similar for the shared compounds. In addition, in both varieties, the relative decrease in γ-terpinene, caryophyllene and humulene emissions following psyllid infestation also appears very similar. By contrast, ‘Micro-Tom’ does not only emit a strikingly smaller number of compounds, but following psyllid infestation α-Pinene emissions, for instance, are as high as in healthy plants in contrast to the ‘Wild’ cultivar, for which α-Pinene emissions appear to decrease (although this decrease was not statistically significant). Therefore, I do not think that domestication explains the observed differences well enough given the similarity between ‘Floradade’ and ‘Wild’ and assuming that ‘Floradade’ and ‘Micro-Tom’ represent domesticated cultivars.

We agree with comments. The majority of compounds presented in figure 1 and 2 were shared in ‘Floradade’ and ‘Wild type’, although in table 1 showed those volatiles emitted exclusively for ‘Wild type’ such as (+)-2-Carene, (-)-α-Pinene and β-Phellandrene.

3) Is there anything known on natural enemies on tomato psyllids when feeding on tomato plants and whether they use olfactory cues to detect their prey/hosts?

Natural enemies reported for tomato psyllids we found parasitoid Tamarixia triozae and Dicyphus hesperus as predator (Ramírez-Ahuja, M de L; Rodríguez-Leyva, E; Lomeli-Flores, JR; Torres-Ruiz, A; Guzmán-Franco, AW. 2017. Evaluating combined use of a parasitoid and a zoophytophagous bug for biological control of the potato psyllid, Bactericera cockerelli. Biological Control 106:9-15. DOI: https://doi.org/10.1016/j.biocontrol.2016.12.003).

Recently, Mayo-Hernández et al. (2019) developed bioassays both in greenhouse (free choice) and laboratory (Y-tube olfactometer) with the same three varieties of tomato with the aim to evaluated parasitism preferences by Tamarixia triozae. Interestingly, T. triozae showed high preference for ‘Wild type’ infested with N3 nymphs compared with ‘Micro-Tom’ and ‘Floradade’ (data unpublished). Although, volatiles organic compounds were not evaluated during the different tritrophic interactions. 

Reviewer 2 Report

There are relatively few studies on tomato chemistry following feeding by the tomato psyllid.  Thus, this study has merit.  I had no specific concerns with the headspace methodology. However, I was surprised that the authors did not use their results to extend the work of Prager et al. 2018 (Prager et al. 2018. Examining the potential role of foliar chemistry in imparting potato germplasm tolerance to potato psyllid, green peach aphid, and zebra chip disease. J. Economic Entomology, 111(1), 2018, 327–336).  Although I believe the paper has merit, I did have some concerns the authors should address prior to publication.

++++++++++++++++++++++++++++++++++++++++++

I struggled with the presentation of the data.

On figures #4 and #5, the authors stated “Boxes with no bars indicate no data were collected for one assay performed under the same conditions”. I think this means that for these figures the data were based on two assays instead of three (if this is not correct then the paper will need to have the methodology rewritten so that the reader can determine what was actually done). In the figures #4 and #5 it was impossible to tell if there were no bars showing variation in specific VOCs because the variation was so small that the bars were not visible, or if all of these bars only had two replicates. In Fig #4, is it correct that only 3 of the 10 compounds tested had three replicates and the rest had only two replicates?  It was not clear to me why, with at least two replicates per VOC, the error bars were not be included. In Fig 3, all compounds were tested in three replicates.I find it quite remarkable that only phenol (VOC #7), among all of the VOCs tested from infested ‘Floradade’ plants, had enough variability to allow the authors to include error bars. If the other VOCs had so little variability that the error bars can not be seen, why are the differences not significant? Some comparisons, such as Fig 4, VOC #8, found comparisons that had significant differences even though others (like #5) had much greater differences but were not significant. Compare the differences between infested and uninfested in #3 versus #5 or #6. Figure 1. If the purpose of this study was to determine if psyllid feeding could account for differences in VOCs, why were the VOC concentration data in Fig 1 not statistically compared? In some cases there were three different plants being compared. The authors apparently only used a t-test for paired samples, which is probably not appropriate for a three sample design. However, it is possible that there was no statistical analysis in the figures that contained the 3-way comparisons.  If this is the case, I recommend the data in all of the figures should be subject to statistical analyses.

The ‘wild type’ tomato needs to be much better defined.There are dozens of ‘wild type’ tomato species.  Providing the location where the wild tomato line (species?) was collected is useful, but 10 years from now the collection site may have changed (multiple tomato species could then be present or the site may have been disturbed to eliminate the species).  A much better approach would be to put some seeds in a repository, or to actually characterize the tomato species tested.  Without this information I am not sure the research on the ‘wild’ line could ever be repeated (which is a basic tenant of science).

Author Response

Reviewer 2:

Comments and Suggestions for Authors

There are relatively few studies on tomato chemistry following feeding by the tomato psyllid.  Thus, this study has merit.  I had no specific concerns with the headspace methodology. However, I was surprised that the authors did not use their results to extend the work of Prager et al. 2018 (Prager et al. 2018. Examining the potential role of foliar chemistry in imparting potato germplasm tolerance to potato psyllid, green peach aphid, and zebra chip disease. J. Economic Entomology, 111(1), 2018, 327–336).  Although I believe the paper has merit, I did have some concerns the authors should address prior to publication.

This was done as suggested (see page 16, L233-236).

++++++++++++++++++++++++++++++++++++++++++

I struggled with the presentation of the data.

On figures #4 and #5, the authors stated “Boxes with no bars indicate no data were collected for one assay performed under the same conditions”. I think this means that for these figures the data were based on two assays instead of three (if this is not correct then the paper will need to have the methodology rewritten so that the reader can determine what was actually done).

Two independent assays were performed and three plants were evaluated per treatment and individual plant was analyzed by SPME and GC-MS. Changes were made as suggested (L276-278 and L280).

In the figures #4 and #5 it was impossible to tell if there were no bars showing variation in specific VOCs because the variation was so small that the bars were not visible, or if all of these bars only had two replicates.

In the figure #4, regarding to healthy plants, compounds 4 and 7 represent one data obtained by GC-MS. For infested plants, compounds 2, 4, 5, 6, 8, and 10 represent one data.

In the figure #5, the compound 2 represent one data obtained by GC-MS for healthy plants. In infested plants, compounds 1 and 2 showed a small error bars (three data).

In Fig #4, is it correct that only 3 of the 10 compounds tested had three replicates and the rest had only two replicates?  It was not clear to me why, with at least two replicates per VOC, the error bars were not be included.

We agree, is correct that only 3 of the 10 compounds tested had three replicates but the rest of compounds had only one data.

In Fig 3, all compounds were tested in three replicates. I find it quite remarkable that only phenol (VOC #7), among all of the VOCs tested from infested ‘Floradade’ plants, had enough variability to allow the authors to include error bars.

For infested plants, the compound 7 had only two data and the rest of compounds had only one data.

If the other VOCs had so little variability that the error bars can not be seen, why are the differences not significant? Some comparisons, such as Fig 4, VOC #8, found comparisons that had significant differences even though others (like #5) had much greater differences but were not significant. Compare the differences between infested and uninfested in #3 versus #5 or #6.

There was an error for compound 8 in infested plants, only one data was obtained and is not comparable. The compound 3 had significant difference but #5 and #6 had only one data.

Figure 1. If the purpose of this study was to determine if psyllid feeding could account for differences in VOCs, why were the VOC concentration data in Fig 1 not statistically compared? In some cases there were three different plants being compared. The authors apparently only used a t-test for paired samples, which is probably not appropriate for a three sample design. However, it is possible that there was no statistical analysis in the figures that contained the 3-way comparisons.  If this is the case, I recommend the data in all of the figures should be subject to statistical analyses.

The statistical analyses were performed in the figure 1, although compounds 9 and 11 had significant differences the rest of compounds were not.

The ‘wild type’ tomato needs to be much better defined. There are dozens of ‘wild type’ tomato species.  Providing the location where the wild tomato line (species?) was collected is useful, but 10 years from now the collection site may have changed (multiple tomato species could then be present or the site may have been disturbed to eliminate the species).  A much better approach would be to put some seeds in a repository, or to actually characterize the tomato species tested.  Without this information I am not sure the research on the ‘wild’ line could ever be repeated (which is a basic tenant of science).

Wild type tomato was collected during summer 2015 in Guanajuato State, Mexico. Wild type tomato was taxonomically characterized as member of Solanum lycopersicum and was deposited with catalogue number XAL0148356 in Herbario XAL of Instituto de Ecología A.C. in Veracruz, Mexico.

Round 2

Reviewer 1 Report

Review of the revised version of “Effects of Bactericera cockerelli herbivory on volatile emissions of three varieties of Solanum lycopersicum” by Mayo-Hernandez et al., submitted to Plants

Overall, I am not satisfied with the changes made and I replied to most of the comments below. The manuscript still needs major editing concerning language and clarity of the text. Also, the methodology is still not described sufficiently. I therefore, cannot recommend to publish the manuscript in its current state. One of the main concerns I have is the interpretation of the data. The abstract says: “We suggest that these qualitative differences in VOC profiles by the degree of domestication could explain the preferences of B. cockerelli.” To me, that seems like an overstatement given the data presented. Assuming a domestication gradient from ‘Wild type’ to ‘Micro-Tom’ and ‘Floradade’ I do not see any representation of this gradient in the VOC emissions. And also VOC emissions did not (obviously, at least) correlate with the host preferences that have been observed earlier. I agree that some of the compounds that were only detected in the ‘Wild type’ might serve as repellent to the psyllids thus putatively explaining the reduced preference for the ‘Wild type’ but this should be discussed in a much clearer way as has been done. Thus, I recommend the authors to revise the respective sections of the manuscript and I also would like to recommend them to perform statistical analyses that are appropriate to answer their research question or, alternatively, rewrite the discussion and be much clearer about the conclusions that are being made and their limitations.

Reviewer 1:

Major comments:

Title:  The title is misleading as is now. I would recommend a rearrangement of the words: “Effects of Bactericera cockerelli herbivory on volatile emissions of three varieties of Solanum lycopersicum”; and also include the term domestication since this is the scope of the manuscript.

We agree with this observation. The title was thereby changed to: “Effects of Bactericera cockerelli herbivory on volatile emissions of three varieties of Solanum lycopersicum”.

Study design:

1) I could only find scarce information on how many replicates were done for each experiment. For the first experiment please state the exact number of for each cultivar in the methods section. For the second experiment the only information I found was that two bioassays were performed. I wonder whether this equals to two replicates. In addition to mentioning this in the methods section, I would wish for this information in the figure captions and also in the tables that are presented.

Changes were made as suggested (L276-278 and L280). Tables and figures captions were also modified.

Please state clearly what the bars and error bars represent, e.g. mean +/- SD. And also state how many replicates these data were obtained from, e.g. N=3. This should be standard.

2) To me the difference between the first experiment on healthy plants and the controls of the second experiment is unclear. As I understood it, in both cases non-infested plants were sampled by SPME for 2.5 hours. This should be clarified. Also, if the “treatments” are the same, were the same compounds detected in the headspace? Because for the second experiment only those are presented that were detected in infested plants.

The table 1 show the VOC profiles identified of healthy plants in three varieties by SPME for 2.5 hours. VOC profile presented in table 2 were identified of infested plants in all varieties for 2.5 hours, although for figures 2, 3, 4 and 5 those compounds that were presented in infested varieties compared with the basal levels in healthy plants (first bioassay).

Then it should be made clear that in the manuscript that healthy plants were only assayed once and that the data shown in figures 3, 4 and 5 are the same as in figure 1. Otherwise, one might think that healthy plants were also included as controls when infested plants were assayed. In fact, that would have been the better experimental design. 

3) How the cultivars were selected is also unclear to me. Here, I would wish for more information on the domestication process of tomato and which species/cultivar is considered the ancestral one; especially for the reader who is not familiar with the topic. In Bergougnoux 2014 (Biotechnology Advances), for instance, the following information can be found:

“The most likely ancestor of cultivated tomatoes is the wild cherry tomato, usually identified as S. lycopersicum var. cerasiforme because of its wide representation in Central America. Nevertheless the genetic investigations made by Nesbitt and Tanksley in 2002 demonstrated that the plants known as “cerasiforme” are a mixture of wild and cultivated tomatoes, rather than the direct ancestor of the cultivated tomato. A very recent study based on the analysis of single nucleotide polymorphisms not only confirms that S. lycopersicum var. cerasiforme is not the ancestor of the cultivated tomato but also reinforces the model that a predomestication of the tomato occurred in the Andean region (Peruvian hypothesis), with the domestication being completed in Mesoamerica (Mexican hypothesis), followed by its introduction to Europe by Spaniards and then spread all over the world (Blanca et al.,2012 ).”

In my opinion, it should be clarified how the used ‘wild-type’ relates to this. Is it also mixture/hybrid of wild and cultivated tomatoes? In addition, it should be mentioned to what extent and why ‘Floradade’ and ‘Micro-Tom’ differ in their degree of domestication. To my (maybe limited) knowledge, both are commercial cultivars and thus the product of extensive breeding. 4) Lastly, I was wondering why an infestation period of only 2 hours was chosen since in many publications that investigated volatile emissions upon infestation by phloem-feeders, and also following tomato psyllid feeding on tomato plants (e.g. Bautista-Lozada et al. 2013 – PLOS One), longer infestation periods of up to several days were chosen.

The cultivars of tomato were selected for availability in our laboratory, ‘Floradade’ (Crown Seeds) was selected as a commercial cultivar commonly used in greenhouse and field conditions. ‘Micro-Tom’ was selected as a cultivar extensively studied in laboratory and greenhouse level, mainly because transgenic and mutants plants were obtained with this background genetic. ‘Wild-type’ was selected according their tolerance to herbivorous insects observed in the environment and was collected with the main objective to evaluate those traits involved in their reduced preference oviposition of Bactericera cockerelli (Mayo Hernández et al. 2018). For us, the degree of domestication in each cultivar of tomato was not determined in our laboratory we expected that ‘Floradade’ is more attractive to pests than ‘Wild-type’ tomato plants.

The bioassays were performed with nymphs of psyllid that harboring the pathogen Candidatus Liberibacter solanacearum (Calso) and 2 hours was chosen for feeding regarding to previous reports in the literature mentioned that Calso could manipulated the plant responses after 24 hours [Sengoda, V.G.; Cooper, W.R.; Swisher, K.D.; Henne, D.C.; Munyaneza, J.E. Latent period and transmission of “Candidatus Liberibacter solanacearum” by the potato psyllid Bactericera cockerelli (Hemiptera: Triozidae). PLoS One. 2014, 9:e93475.]

Thanks for the explanation. Please also include some more background on the domestication history of tomato in the introduction. I still wonder in what way ‘Floradade’ and ‘Micro-Tom’ differ in their degree of domestication. Even though, you did not determine this in your lab, you should provide some more information since this is crucial to your hypotheses and conclusion (see below).

Material and methods:

1) As mentioned above, I miss information on the exact study design. How many replicates were done? What were the differences between the healthy plants of experiment 1 and experiment 2? 

Changes were made as suggested (L276-278 and L280).

Thanks for the information. There are still some uncertainties to me, though. And this should also be much clearer in the methods sections. Again, I would wish for the precise number of replicates and how mean values were calculated then.

As far as I understand, you performed one bioassay for the first experiment on healthy plants. First, I do not understand what you mean in the figure caption of figure 1: “[…] no bars indicate no data were collected for one assay […]” (ll.115), since only one assays was done. Second, to me, each assayed plant would represent a replicate, thus resulting in three replicates per genotype. I assume, that if no error bars are shown in a given figure, this means that the respective compound was only detected in one sample. What happens if you only consider those volatiles that were detected in the headspace of at least two out of three plants per genotype? Such a filtering is commonly seen in similar studies and might result in a much more robust data set. (This also applies to the infested plants)

For infested plants, two bioassays were done. Does this equal to a maximum of 2x3 replicates? If the two bioassays did not differ strongly, i.e. you see the same emissions, would it not be possible to calculate mean values and standard deviations across all these replicates; and also do the statistical analyses across all the replicates? If this was possible, you could do some more appropriate statistical analyses (see below).

With regard to ll.276-278: Did you assay each individual plant three times as written there resulting in three GC-MS runs per individual? How was that data treated? Did you calculate mean peak areas for each individual plant before calculating the mean emission across the three replicates? Please clarify!

2) Analytics: I miss information on how the individual compounds were identified? Which library were the mass spectra compared to? Were the retention times and mass spectra compared to an in-house library or authenticated standards? Were Kovats indices determined for the individual compounds to assist identification? Please clarify! 

All the compounds were identified by a NIST92 and Wiley library, according the percentage presented (>90%), retentions times were compared with library and Kovats indices also were determined for individual compounds.

Please include this information in the methods sections.

3) Related to the previous point: the authors might include a table in the supporting information where the retention times, Kovats indices, if determined, and major m/z for each compound are reported; alongside with an exemplary and labelled chromatogram.

We agree with comment, but we considered that the information in tables could be enough to explain those differences between varieties of tomato.

This table is not meant to interpret your findings but rather to be able to fully retrace what has been done.

Results and data analysis:

In general, I would recommend to structure the results section using appropriate sub headings. For this manuscript this might just be two, one for constitutive emissions and one for induced emissions. Nevertheless, structure will help the reader to capture the main messages better. In the current form, the results go from constitutively released volatiles to induced emissions and back to constitutive emissions.

Changes were made as recommended.

Thank you for including the sub headings. However, in ll. 136-138 you compare VOC emissions from healthy plants under the sub heading: “Volatile organic compounds released in infested plants with Bactericera cockerelli.” This should be rearranged accordingly.

1) Please number all compounds consecutively from 1 to n across tables 1 and 2 and stick to that numbering in all figures. This will make it a lot easier for the reader to compare between figures and tables.

Changes were made as suggested.

This has not been done appropriately. I meant that you would number the compounds starting from the first compound in table 1 to the last compound in table 2. Like this, beta-Pinene should have the same number in both tables. Also, this would facilitate an easier comparison between the different figures, since a given compound would then also have the same number throughout the manuscript.

2) As a recommendation: since all peak area units given for the individual compounds are arbitrary anyway, and range across app. four orders of magnitude, the authors might as well just use numbers in the range from 0.1 to 100. In my opinion, this is much more intuitive to read and will also be useful for the next point.

We selected the symbols (+, ++…etc.) in order to explain the possible differences in abundance between varieties for table 1 and 2. Also, as mentioned above some compounds were detected only once.

Okay. If you stick with the symbols, you might include a table with the actual numbers in the supporting information then.

3) Concerning the two tables, I would prefer to see actual numbers, e.g. mean±SD, instead of “+”, “++” etc. Concerning the data presentation, I also have a few suggestions:

4) Table 2: I would recommend to design the table in the following format

 RT Wild Floradade Micro-Tom Compound (min) healthy infested healthy infested healthy Infested 1 α-Pinene 4.574 25±5 22±4 etc…    and give mean peak areas ± SD.

Please, see above comment.

You might consider this table to be included in the supporting information then.

5) Tables 1 and 2: Please include a line in the end giving the total emissions for each column. 

Changes were made as recommended.

This has also not be done appropriately. If you prepare a table with the actual peak areas please also include the total peak area at the bottom line as an estimate for total VOC emission.

6) Was it considered to perform a multivariate analysis on the observed volatile profiles of healthy plants, e.g. a principal component analysis (PCA) or a partial least squares discriminant analysis (PLS-DA, e.g. see Fatouros et al. 2012 – PLOS One; https://journals.plos.org/plosone/article?id=10.1371/journal.pone.0043607) instead of showing the measured emissions for a selection of 11 compounds for the three cultivars (Figure 1)? This might yield some more insight whether the profiles actually differ between the three cultivars. Assuming, of course, that a valid model can be built with the number of replicates at hand.  In my opinion, a composite figure showing the total emissions, including an appropriate statistical test, and the outcome of the PCA would then give some good idea in how far the different cultivars actually differ and whether the domestication gradient is represented in the volatile profiles. The same applies to the second experiment. Although, as it seems to me for the moment, only two replicates have been done, which will not suffice for this kind of analysis.

We agree with comments, in different compounds either in healthy and infested plants had not minimal replicates obtained for statistical analysis. Regarding the reference (Fatouros et al. 2012), they evaluated volatile emissions measured as peak area divided by the fresh weight of a plant’s foliage using the software SIMCA P+12.0, however, in our bioassays performed we did not weight the fresh foliar tissue.

Nevertheless, would it not be possible to perform a PCA on your peak areas? Which data you use as an input for the analysis was not the point here.

7) Concerning the number of replicates: I wonder how t-tests were done for the data shown in figures 4 and 5? It is stated that “boxes with no bars indicate no data were collected for one assay”. And there are quite a few boxes without bars suggesting that there was only one replicate. With this low number of replicates a non-parametric test might be more appropriate, anyway.

We agree with comment. The figures were corrected and error bars were removed to avoid confusion in replicates number.

8) Concerning figures 3 to 5: How were the depicted compounds selected? Since only 11, 12, and 4 compounds were reported for infested ‘Wild’, ‘Floradade’ and ‘Micro-Tom’, respectively, in table 2, would it be possible to show all of them in the figures?

Total compounds were selected for their abundance with >90% presented after GC-MS analysis.

Theoretical background & data interpretation:

1) In ll. 200 it is stated: “With the qualitative differences among varieties of tomato VOC we speculated that some volatiles could act as attractants or repellents for B. cockerelli adults, although in this study we did not perform electroantennography analysis to validate the results.” First of all, it is not mentioned which compounds were speculated on as attractants

or repellents. Is there anything known on individual compounds that are attractive or repellent to B. cockerelli? Would it be possible to test whether B. cockerelli adults display preferences for any of the cultivars used here? Could the authors construct a simple y-tube olfactometer (e.g. as in Fatouros et al. 2012 – PLOS One, Figure 5) to test this? In this kind of setup maybe even individual compounds could be tested. While these assays are not strictly required for the present manuscript, they would considerably improve it and strengthen the hypothesis that the host plant choices that have been observed earlier are indeed mediated by contrasting volatile emissions. Especially so, since the host plant preferences that are referred to were observed during cage experiments where the insects can directly access the plants and volatiles may thus only play a minor role for host plant acceptance. 

We speculated that (+)-4-carene and β-phellandrene could act as repellent in wild plants against to B. cockerelli, although we considered in future experiments to found by Y-tube olfactometer and electroantennography analysis those compounds involved either in repellence or attraction. Regarding to repellent compounds, we found dimethyl sulfide (DMDS) and five plant essential oils (thyme, tea tree, peppermint, savory and clove), DMDS and plant oils had a significant repellent effect on adults of B. cockerelli, however, individual compounds were not determined in this study (John Diaz-Montano and John T. Trumble. 2013. Behavioral responses of the potato psyllid (Hemiptera: Triozidae) to volatiles from dimethyl sulfide and plant essential oils. J Insect Behav 26:336-351). Recently, Mayo-Hernández et al. (2018) reported that high preference of oviposition by B. cockerelli was presented in ‘Floradade’ and ‘Wild type’ showed a lower preferences (Mayo-Hernández, J.; Flores-Olivas, A.; Valenzuela-Soto, J.H.; Rodríguez-Pagaza, Y.; Vega-Chávez, J.; Hernández-Castillo, F.; Aguirre-Uribe, L. Bactericera cockerelli Sulc oviposition preference and development on three tomato varieties. Southwest Entomol. 2018, 43, 905–910).

2) Overall, ‘Floradade’ and ‘Wild type’ appear very similar to me, both in their constitutive emissions but also considering the volatile emissions following psyllid infestation. As shown in Figures 1 and 2 the released amounts are very similar for the shared compounds. In addition, in both varieties, the relative decrease in γ-terpinene, caryophyllene and humulene emissions following psyllid infestation also appears very similar. By contrast, ‘Micro-Tom’ does not only emit a strikingly smaller number of compounds, but following psyllid infestation α-Pinene emissions, for instance, are as high as in healthy plants in contrast to the ‘Wild’ cultivar, for which α-Pinene emissions appear to decrease (although this decrease was not statistically significant). Therefore, I do not think that domestication explains the observed differences well enough given the similarity between ‘Floradade’ and ‘Wild’ and assuming that ‘Floradade’ and ‘Micro-Tom’ represent domesticated cultivars.

We agree with comments. The majority of compounds presented in figure 1 and 2 were shared in ‘Floradade’ and ‘Wild type’, although in table 1 showed those volatiles emitted exclusively for ‘Wild type’ such as (+)-2-Carene, (-)-α-Pinene and β-Phellandrene.

I will just try to summarise the present study for myself in order to get things right.

You previously observed that psyllids show differential preferences for ‘Floradade’, ‘Micro-Tom’ and ‘Wild type’ in terms of eggs being laid (Mayo-Hernandez et al. 2018). For the current manuscript you therefore reasoned that (I) these differences might be explained by differences in volatile emissions and (II) that differences in volatile emissions between these three varieties could be attributed to their differing degrees of domestication.

For both hypotheses, I miss the appropriate statistical analyses.

Regarding the first hypothesis, I still think that a multivariate approach, e.g. a PCA, would be appropriate to observe similarities/dissimilarities between the different genotypes. This could be applied to both constitutive AND induced emissions.

Regarding the second hypotheses, you might do a multi-factorial ANOVA including treatment (healthy/infested) and genotype (‘Floradade’, ’Micro-Tom’ and ‘Wild type’) as factors; although the treatments/bioassays have not been done at the same time. This would indicate to what extent the three different cultivars differ.

But, of course, this can only be done with a sufficient amount of replication (see above).

If these tests cannot be done due to missing replication then all of the conclusions drawn have to be done so in a much more careful way.

What I think, one can conclude on the data as it is currently being presented is that: (I) volatile emissions from healthy plants cannot explain the previously observed differences in host preferences, although healthy ‘Wild type’ plants emitted (+)-2-carene, (-)-alpha-pinene, beta-phellandrene in contrast to the other two genotypes. As mentioned above, it would be interesting to test the repellency of these compounds in olfactometer assays. Admittedly, this is being discussed but I had to go through your discussion several times to find this. Why not speculate about these compounds where you literally mention that you “speculated that some volatiles could act as attractants and repellents […]” (ll. 201)? And with regard to the second hypothesis, I think it is safe to conclude that the differences in volatile emissions, also after psyllid feeding, cannot be attributed to a degree in domestication since, as mentioned above, ‘Floradade’ and ‘Wild type’ seem to be more similar than ‘Floradade’ and ‘Micro-Tom’.

3) Is there anything known on natural enemies on tomato psyllids when feeding on tomato plants and whether they use olfactory cues to detect their prey/hosts?

Natural enemies reported for tomato psyllids we found parasitoid Tamarixia triozae and Dicyphus hesperus as predator (Ramírez-Ahuja, M de L; Rodríguez-Leyva, E; Lomeli-Flores, JR; Torres-Ruiz, A; Guzmán-Franco, AW. 2017. Evaluating combined use of a parasitoid and a zoophytophagous bug for biological control of the potato psyllid, Bactericera cockerelli. Biological Control 106:9-15. DOI: https://doi.org/10.1016/j.biocontrol.2016.12.003).

Recently, Mayo-Hernández et al. (2019) developed bioassays both in greenhouse (free choice) and laboratory (Y-tube olfactometer) with the same three varieties of tomato with the aim to evaluated parasitism preferences by Tamarixia triozae. Interestingly, T. triozae showed high preference for ‘Wild type’ infested with N3 nymphs compared with ‘Micro-Tom’ and ‘Floradade’ (data unpublished). Although, volatiles organic compounds were not evaluated during the different tritrophic interactions. 

Thanks for the information. That sounds very interesting. I wonder whether you might want to wait until the part on predator preferences is finished, and, if possible, try to quantify volatile emissions in parallel. This would strengthen your manuscript a lot.

Author Response

Review of the revised version of “Effects of Bactericera cockerelli herbivory on volatile emissions of three varieties of Solanum lycopersicum” by Mayo-Hernandez et al., submitted to Plants

Overall, I am not satisfied with the changes made and I replied to most of the comments below. The manuscript still needs major editing concerning language and clarity of the text. Also, the methodology is still not described sufficiently. I therefore, cannot recommend to publish the manuscript in its current state. One of the main concerns I have is the interpretation of the data. The abstract says: “We suggest that these qualitative differences in VOC profiles by the degree of domestication could explain the preferences of B. cockerelli.” To me, that seems like an overstatement given the data presented. Assuming a domestication gradient from ‘Wild type’ to ‘Micro-Tom’ and ‘Floradade’ I do not see any representation of this gradient in the VOC emissions. And also VOC emissions did not (obviously, at least) correlate with the host preferences that have been observed earlier. I agree that some of the compounds that were only detected in the ‘Wild type’ might serve as repellent to the psyllids thus putatively explaining the reduced preference for the ‘Wild type’ but this should be discussed in a much clearer way as has been done. Thus, I recommend the authors to revise the respective sections of the manuscript and I also would like to recommend them to perform statistical analyses that are appropriate to answer their research question or, alternatively, rewrite the discussion and be much clearer about the conclusions that are being made and their limitations.

Reviewer 1:

Major comments:

Title:  The title is misleading as is now. I would recommend a rearrangement of the words: “Effects of Bactericera cockerelli herbivory on volatile emissions of three varieties of Solanum lycopersicum”; and also include the term domestication since this is the scope of the manuscript.

We agree with this observation. The title was thereby changed to: “Effects of Bactericera cockerelli herbivory on volatile emissions of three varieties of Solanum lycopersicum”.

Study design:

1) I could only find scarce information on how many replicates were done for each experiment. For the first experiment please state the exact number of for each cultivar in the methods section. For the second experiment the only information I found was that two bioassays were performed. I wonder whether this equals to two replicates. In addition to mentioning this in the methods section, I would wish for this information in the figure captions and also in the tables that are presented.

Changes were made as suggested (L276-278 and L280). Tables and figures captions were also modified.

Please state clearly what the bars and error bars represent, e.g. mean +/- SD. And also state how many replicates these data were obtained from, e.g. N=3. This should be standard.

R2: Changes were done as requested.

2) To me the difference between the first experiment on healthy plants and the controls of the second experiment is unclear. As I understood it, in both cases non-infested plants were sampled by SPME for 2.5 hours. This should be clarified. Also, if the “treatments” are the same, were the same compounds detected in the headspace? Because for the second experiment only those are presented that were detected in infested plants.

The table 1 show the VOC profiles identified of healthy plants in three varieties by SPME for 2.5 hours. VOC profile presented in table 2 were identified of infested plants in all varieties for 2.5 hours, although for figures 2, 3, 4 and 5 those compounds that were presented in infested varieties compared with the basal levels in healthy plants (first bioassay).

Then it should be made clear that in the manuscript that healthy plants were only assayed once and that the data shown in figures 3, 4 and 5 are the same as in figure 1. Otherwise, one might think that healthy plants were also included as controls when infested plants were assayed. In fact, that would have been the better experimental design. 

R2: Figures 3, 4 and 5 were removed in the manuscript and we included figures 1 and 2 (re-structured).

3) How the cultivars were selected is also unclear to me. Here, I would wish for more information on the domestication process of tomato and which species/cultivar is considered the ancestral one; especially for the reader who is not familiar with the topic. In Bergougnoux 2014 (Biotechnology Advances), for instance, the following information can be found:

“The most likely ancestor of cultivated tomatoes is the wild cherry tomato, usually identified as S. lycopersicum var. cerasiforme because of its wide representation in Central America. Nevertheless the genetic investigations made by Nesbitt and Tanksley in 2002 demonstrated that the plants known as “cerasiforme” are a mixture of wild and cultivated tomatoes, rather than the direct ancestor of the cultivated tomato. A very recent study based on the analysis of single nucleotide polymorphisms not only confirms that S. lycopersicum var. cerasiforme is not the ancestor of the cultivated tomato but also reinforces the model that a predomestication of the tomato occurred in the Andean region (Peruvian hypothesis), with the domestication being completed in Mesoamerica (Mexican hypothesis), followed by its introduction to Europe by Spaniards and then spread all over the world (Blanca et al.,2012 ).”

In my opinion, it should be clarified how the used ‘wild-type’ relates to this. Is it also mixture/hybrid of wild and cultivated tomatoes? In addition, it should be mentioned to what extent and why ‘Floradade’ and ‘Micro-Tom’ differ in their degree of domestication. To my (maybe limited) knowledge, both are commercial cultivars and thus the product of extensive breeding. 4) Lastly, I was wondering why an infestation period of only 2 hours was chosen since in many publications that investigated volatile emissions upon infestation by phloem-feeders, and also following tomato psyllid feeding on tomato plants (e.g. Bautista-Lozada et al. 2013 – PLOS One), longer infestation periods of up to several days were chosen.

The cultivars of tomato were selected for availability in our laboratory, ‘Floradade’ (Crown Seeds) was selected as a commercial cultivar commonly used in greenhouse and field conditions. ‘Micro-Tom’ was selected as a cultivar extensively studied in laboratory and greenhouse level, mainly because transgenic and mutants plants were obtained with this background genetic. ‘Wild-type’ was selected according their tolerance to herbivorous insects observed in the environment and was collected with the main objective to evaluate those traits involved in their reduced preference oviposition of Bactericera cockerelli (Mayo Hernández et al. 2018). For us, the degree of domestication in each cultivar of tomato was not determined in our laboratory we expected that ‘Floradade’ is more attractive to pests than ‘Wild-type’ tomato plants.

The bioassays were performed with nymphs of psyllid that harboring the pathogen Candidatus Liberibacter solanacearum (Calso) and 2 hours was chosen for feeding regarding to previous reports in the literature mentioned that Calso could manipulated the plant responses after 24 hours [Sengoda, V.G.; Cooper, W.R.; Swisher, K.D.; Henne, D.C.; Munyaneza, J.E. Latent period and transmission of “Candidatus Liberibacter solanacearum” by the potato psyllid Bactericera cockerelli (Hemiptera: Triozidae). PLoS One. 2014, 9:e93475.]

Thanks for the explanation. Please also include some more background on the domestication history of tomato in the introduction. I still wonder in what way ‘Floradade’ and ‘Micro-Tom’ differ in their degree of domestication. Even though, you did not determine this in your lab, you should provide some more information since this is crucial to your hypotheses and conclusion (see below).

R2: The Floradade and Micro-Tom differ in genomic size. Micro-Tom has a tiny genome, is a model organism for their short life cycle, and small size. Micro-tom presents a genetic difference with other commercial tomato cultivars.

Shirasawa, K., Isobe, S., Hirakawa, H., Asamizu, E., Fukuoka, H., Just, D. et al. (2010) SNP discovery and linkage map construction in cultivated tomato. DNA Res. 17: 381–391

Hirakawa, H., Shirasawa, K., Ohyama, A., Fukuoka, H., Aoki, K., Rothan, C. et al. (2013) Genome-wide SNP genotyping to infer the effects on gene functions in tomato. DNA Res. 20: 221–233.

Material and methods:

1) As mentioned above, I miss information on the exact study design. How many replicates were done? What were the differences between the healthy plants of experiment 1 and experiment 2? 

Changes were made as suggested (L276-278 and L280).

Thanks for the information. There are still some uncertainties to me, though. And this should also be much clearer in the methods sections. Again, I would wish for the precise number of replicates and how mean values were calculated then.

R: Changes are made as suggested.

As far as I understand, you performed one bioassay for the first experiment on healthy plants. First, I do not understand what you mean in the figure caption of figure 1: “[…] no bars indicate no data were collected for one assay […]” (ll.115), since only one assays was done. Second, to me, each assayed plant would represent a replicate, thus resulting in three replicates per genotype. I assume, that if no error bars are shown in a given figure, this means that the respective compound was only detected in one sample. What happens if you only consider those volatiles that were detected in the headspace of at least two out of three plants per genotype? Such a filtering is commonly seen in similar studies and might result in a much more robust data set. (This also applies to the infested plants)

R: Changes were made as recommended.

For infested plants, two bioassays were done. Does this equal to a maximum of 2x3 replicates? If the two bioassays did not differ strongly, i.e. you see the same emissions, would it not be possible to calculate mean values and standard deviations across all these replicates; and also do the statistical analyses across all the replicates? If this was possible, you could do some more appropriate statistical analyses (see below).

R: Changes were done as requested.

With regard to ll.276-278: Did you assay each individual plant three times as written there resulting in three GC-MS runs per individual? How was that data treated? Did you calculate mean peak areas for each individual plant before calculating the mean emission across the three replicates? Please clarify!

R: We regret this mistake and was corrected in the manuscript.

2) Analytics: I miss information on how the individual compounds were identified? Which library were the mass spectra compared to? Were the retention times and mass spectra compared to an in-house library or authenticated standards? Were Kovats indices determined for the individual compounds to assist identification? Please clarify! 

All the compounds were identified by a NIST92 and Wiley library, according the percentage presented (>90%), retentions times were compared with library and Kovats indices also were determined for individual compounds.

Please include this information in the methods sections.

R: Changes are made as suggested.

3) Related to the previous point: the authors might include a table in the supporting information where the retention times, Kovats indices, if determined, and major m/z for each compound are reported; alongside with an exemplary and labelled chromatogram.

We agree with comment, but we considered that the information in tables could be enough to explain those differences between varieties of tomato.

This table is not meant to interpret your findings but rather to be able to fully retrace what has been done.

Results and data analysis:

In general, I would recommend to structure the results section using appropriate sub headings. For this manuscript this might just be two, one for constitutive emissions and one for induced emissions. Nevertheless, structure will help the reader to capture the main messages better. In the current form, the results go from constitutively released volatiles to induced emissions and back to constitutive emissions.

Changes were made as recommended.

Thank you for including the sub headings. However, in ll. 136-138 you compare VOC emissions from healthy plants under the sub heading: “Volatile organic compounds released in infested plants with Bactericera cockerelli.” This should be rearranged accordingly.

R: Changes were done as recommended.

1) Please number all compounds consecutively from 1 to n across tables 1 and 2 and stick to that numbering in all figures. This will make it a lot easier for the reader to compare between figures and tables.

Changes were made as suggested.

This has not been done appropriately. I meant that you would number the compounds starting from the first compound in table 1 to the last compound in table 2. Like this, beta-Pinene should have the same number in both tables. Also, this would facilitate an easier comparison between the different figures, since a given compound would then also have the same number throughout the manuscript.

R: The tables 1 and 2 were changed as requested (re-structured).

2) As a recommendation: since all peak area units given for the individual compounds are arbitrary anyway, and range across app. four orders of magnitude, the authors might as well just use numbers in the range from 0.1 to 100. In my opinion, this is much more intuitive to read and will also be useful for the next point.

We selected the symbols (+, ++…etc.) in order to explain the possible differences in abundance between varieties for table 1 and 2. Also, as mentioned above some compounds were detected only once.

Okay. If you stick with the symbols, you might include a table with the actual numbers in the supporting information then.

R: The tables 1 and 2 were changed as requested (re-structured).

3) Concerning the two tables, I would prefer to see actual numbers, e.g. mean±SD, instead of “+”, “++” etc. Concerning the data presentation, I also have a few suggestions:

4) Table 2: I would recommend to design the table in the following format

RT Wild Floradade Micro-Tom Compound (min) healthy infested healthy infested healthy Infested 1 α-Pinene 4.574 25±5 22±4 etc…    and give mean peak areas ± SD.

Please, see above comment.

You might consider this table to be included in the supporting information then.

R: The tables 1 and 2 were changed as requested (re-structured).

5) Tables 1 and 2: Please include a line in the end giving the total emissions for each column. 

Changes were made as recommended.

This has also not be done appropriately. If you prepare a table with the actual peak areas please also include the total peak area at the bottom line as an estimate for total VOC emission.

R: The tables were re-structured; the raw data of peak areas were log transformed.

6) Was it considered to perform a multivariate analysis on the observed volatile profiles of healthy plants, e.g. a principal component analysis (PCA) or a partial least squares discriminant analysis (PLS-DA, e.g. see Fatouros et al. 2012 – PLOS One; https://journals.plos.org/plosone/article?id=10.1371/journal.pone.0043607) instead of showing the measured emissions for a selection of 11 compounds for the three cultivars (Figure 1)? This might yield some more insight whether the profiles actually differ between the three cultivars. Assuming, of course, that a valid model can be built with the number of replicates at hand.  In my opinion, a composite figure showing the total emissions, including an appropriate statistical test, and the outcome of the PCA would then give some good idea in how far the different cultivars actually differ and whether the domestication gradient is represented in the volatile profiles. The same applies to the second experiment. Although, as it seems to me for the moment, only two replicates have been done, which will not suffice for this kind of analysis.

We agree with comments, in different compounds either in healthy and infested plants had not minimal replicates obtained for statistical analysis. Regarding the reference (Fatouros et al. 2012), they evaluated volatile emissions measured as peak area divided by the fresh weight of a plant’s foliage using the software SIMCA P+12.0, however, in our bioassays performed we did not weight the fresh foliar tissue.

Nevertheless, would it not be possible to perform a PCA on your peak areas? Which data you use as an input for the analysis was not the point here.

R: Unfortunately for us, it was not always possible to obtain the additional information required.

7) Concerning the number of replicates: I wonder how t-tests were done for the data shown in figures 4 and 5? It is stated that “boxes with no bars indicate no data were collected for one assay”. And there are quite a few boxes without bars suggesting that there was only one replicate. With this low number of replicates a non-parametric test might be more appropriate, anyway.

We agree with comment. The figures were corrected and error bars were removed to avoid confusion in replicates number.

8) Concerning figures 3 to 5: How were the depicted compounds selected? Since only 11, 12, and 4 compounds were reported for infested ‘Wild’, ‘Floradade’ and ‘Micro-Tom’, respectively, in table 2, would it be possible to show all of them in the figures?

Total compounds were selected for their abundance with >90% presented after GC-MS analysis.

Theoretical background & data interpretation:

1) In ll. 200 it is stated: “With the qualitative differences among varieties of tomato VOC we speculated that some volatiles could act as attractants or repellents for B. cockerelli adults, although in this study we did not perform electroantennography analysis to validate the results.” First of all, it is not mentioned which compounds were speculated on as attractants

or repellents. Is there anything known on individual compounds that are attractive or repellent to B. cockerelli? Would it be possible to test whether B. cockerelli adults display preferences for any of the cultivars used here? Could the authors construct a simple y-tube olfactometer (e.g. as in Fatouros et al. 2012 – PLOS One, Figure 5) to test this? In this kind of setup maybe even individual compounds could be tested. While these assays are not strictly required for the present manuscript, they would considerably improve it and strengthen the hypothesis that the host plant choices that have been observed earlier are indeed mediated by contrasting volatile emissions. Especially so, since the host plant preferences that are referred to were observed during cage experiments where the insects can directly access the plants and volatiles may thus only play a minor role for host plant acceptance. 

We speculated that (+)-4-carene and β-phellandrene could act as repellent in wild plants against to B. cockerelli, although we considered in future experiments to found by Y-tube olfactometer and electroantennography analysis those compounds involved either in repellence or attraction. Regarding to repellent compounds, we found dimethyl sulfide (DMDS) and five plant essential oils (thyme, tea tree, peppermint, savory and clove), DMDS and plant oils had a significant repellent effect on adults of B. cockerelli, however, individual compounds were not determined in this study (John Diaz-Montano and John T. Trumble. 2013. Behavioral responses of the potato psyllid (Hemiptera: Triozidae) to volatiles from dimethyl sulfide and plant essential oils. J Insect Behav 26:336-351). Recently, Mayo-Hernández et al. (2018) reported that high preference of oviposition by B. cockerelli was presented in ‘Floradade’ and ‘Wild type’ showed a lower preferences (Mayo-Hernández, J.; Flores-Olivas, A.; Valenzuela-Soto, J.H.; Rodríguez-Pagaza, Y.; Vega-Chávez, J.; Hernández-Castillo, F.; Aguirre-Uribe, L. Bactericera cockerelli Sulc oviposition preference and development on three tomato varieties. Southwest Entomol. 2018, 43, 905–910).

2) Overall, ‘Floradade’ and ‘Wild type’ appear very similar to me, both in their constitutive emissions but also considering the volatile emissions following psyllid infestation. As shown in Figures 1 and 2 the released amounts are very similar for the shared compounds. In addition, in both varieties, the relative decrease in γ-terpinene, caryophyllene and humulene emissions following psyllid infestation also appears very similar. By contrast, ‘Micro-Tom’ does not only emit a strikingly smaller number of compounds, but following psyllid infestation α-Pinene emissions, for instance, are as high as in healthy plants in contrast to the ‘Wild’ cultivar, for which α-Pinene emissions appear to decrease (although this decrease was not statistically significant). Therefore, I do not think that domestication explains the observed differences well enough given the similarity between ‘Floradade’ and ‘Wild’ and assuming that ‘Floradade’ and ‘Micro-Tom’ represent domesticated cultivars.

We agree with comments. The majority of compounds presented in figure 1 and 2 were shared in ‘Floradade’ and ‘Wild type’, although in table 1 showed those volatiles emitted exclusively for ‘Wild type’ such as (+)-2-Carene, (-)-α-Pinene and β-Phellandrene.

I will just try to summarise the present study for myself in order to get things right.

You previously observed that psyllids show differential preferences for ‘Floradade’, ‘Micro-Tom’ and ‘Wild type’ in terms of eggs being laid (Mayo-Hernandez et al. 2018). For the current manuscript you therefore reasoned that (I) these differences might be explained by differences in volatile emissions and (II) that differences in volatile emissions between these three varieties could be attributed to their differing degrees of domestication.

For both hypotheses, I miss the appropriate statistical analyses.

Regarding the first hypothesis, I still think that a multivariate approach, e.g. a PCA, would be appropriate to observe similarities/dissimilarities between the different genotypes. This could be applied to both constitutive AND induced emissions.

Regarding the second hypotheses, you might do a multi-factorial ANOVA including treatment (healthy/infested) and genotype (‘Floradade’, ’Micro-Tom’ and ‘Wild type’) as factors; although the treatments/bioassays have not been done at the same time. This would indicate to what extent the three different cultivars differ.

But, of course, this can only be done with a sufficient amount of replication (see above).

If these tests cannot be done due to missing replication then all of the conclusions drawn have to be done so in a much more careful way.

What I think, one can conclude on the data as it is currently being presented is that: (I) volatile emissions from healthy plants cannot explain the previously observed differences in host preferences, although healthy ‘Wild type’ plants emitted (+)-2-carene, (-)-alpha-pinene, beta-phellandrene in contrast to the other two genotypes. As mentioned above, it would be interesting to test the repellency of these compounds in olfactometer assays. Admittedly, this is being discussed but I had to go through your discussion several times to find this. Why not speculate about these compounds where you literally mention that you “speculated that some volatiles could act as attractants and repellents […]” (ll. 201)? And with regard to the second hypothesis, I think it is safe to conclude that the differences in volatile emissions, also after psyllid feeding, cannot be attributed to a degree in domestication since, as mentioned above, ‘Floradade’ and ‘Wild type’ seem to be more similar than ‘Floradade’ and ‘Micro-Tom’.

R: The table 1 include the all VOC profiles in different varieties, we found some qualitative differences in compounds detected and we speculate that this differences could influence the oviposition preferences by tomato-psyllid (according Mayo-Hernandez et al., 2018).

3) Is there anything known on natural enemies on tomato psyllids when feeding on tomato plants and whether they use olfactory cues to detect their prey/hosts?

Natural enemies reported for tomato psyllids we found parasitoid Tamarixia triozae and Dicyphus hesperus as predator (Ramírez-Ahuja, M de L; Rodríguez-Leyva, E; Lomeli-Flores, JR; Torres-Ruiz, A; Guzmán-Franco, AW. 2017. Evaluating combined use of a parasitoid and a zoophytophagous bug for biological control of the potato psyllid, Bactericera cockerelli. Biological Control 106:9-15. DOI: https://doi.org/10.1016/j.biocontrol.2016.12.003).

Recently, Mayo-Hernández et al. (2019) developed bioassays both in greenhouse (free choice) and laboratory (Y-tube olfactometer) with the same three varieties of tomato with the aim to evaluated parasitism preferences by Tamarixia triozae. Interestingly, T. triozae showed high preference for ‘Wild type’ infested with N3 nymphs compared with ‘Micro-Tom’ and ‘Floradade’ (data unpublished). Although, volatiles organic compounds were not evaluated during the different tritrophic interactions. 

Thanks for the information. That sounds very interesting. I wonder whether you might want to wait until the part on predator preferences is finished, and, if possible, try to quantify volatile emissions in parallel. This would strengthen your manuscript a lot.

R: We agree with this observation and we will consider for next manuscript.

Reviewer 2 Report

I am concerned that the authors do not have enough data to publish the comparisons  (fig 4 is used as an example).  In their response to my questions regarding how many replicates were used, they added the statement to the figure  that: "Two bioassays were performed independently with three biological replicates."

However, in their response to the review they stated: "In the figure #4, regarding to healthy plants, compounds 4 and 7 represent one data obtained by GC-MS. For infested plants, compounds 2, 4, 5, 6, 8, and 10 represent one data".  Does "one data" mean that there was one replicate where  three plants were individually tested or that there was only one plant tested?  If the authors tested only one plant, and there is therefore no variability, then the paper does not have enough information to justify publication.  If they tested 3 replicate plants then why not use the variability inherent in the three individual plant samples to conduct a non-parametric statistical analysis to compare the amounts of the various VOCs?  I do not believe it is reasonable to make the reader try to determine if the bars in the figures have either 1) very small error bars that did not reproduce well or 2) no error bars because there was no replication.   

Author Response

Reviewer 2:

Comments and Suggestions for Authors

I am concerned that the authors do not have enough data to publish the comparisons  (fig 4 is used as an example).  In their response to my questions regarding how many replicates were used, they added the statement to the figure  that: "Two bioassays were performed independently with three biological replicates."

R: Changes were made as requested.

However, in their response to the review they stated: "In the figure #4, regarding to healthy plants, compounds 4 and 7 represent one data obtained by GC-MS. For infested plants, compounds 2, 4, 5, 6, 8, and 10 represent one data".  Does "one data" mean that there was one replicate where  three plants were individually tested or that there was only one plant tested?

R: We regret this mistake and was corrected in the manuscript.

If the authors tested only one plant, and there is therefore no variability, then the paper does not have enough information to justify publication.  If they tested 3 replicate plants then why not use the variability inherent in the three individual plant samples to conduct a non-parametric statistical analysis to compare the amounts of the various VOCs?  

R: The statistical analysis was made as requested.

I do not believe it is reasonable to make the reader try to determine if the bars in the figures have either 1) very small error bars that did not reproduce well or 2) no error bars because there was no replication.   

R: Changes were made as requested.

Submission Date

08 August 2019

Date of this review

06 Sep 2019 19:22:15

Round 3

Reviewer 1 Report

Comments on revised version of Mayo-Hernandez et al. – “Effects of Bactericera cockerelli herbivory on Voltaile Emissions of Three Varieties of Solanum lycopersicum (plants-580200_REV2_JHVS)

I admit that considerable improvements have been made. Language and grammar still need a revision, though. I will just mention a few examples:

l 36: “we found…”; please avoid subjective language.

ll45: “…that domesticated tomato reduced the attraction or…”   imprecise language. Rather, the attraction of predators was reduced during the domestication process, or: domestication resulted in reduced predator attraction as compared to wild tomato varieties, etc…

l65: “markedly in the blends emitted by plants evaluated”

l197: “we found”

ll227: “This report confirms that tomato varieties respond in different ways to psyllids infestation at different life stages.” This sentence implies that you examined whether tomato responses to psyllid infestation depend on psyllid life stage in the present study. But this is not the case.

Overall, Introduction and Material & methods seem fine to me. I only have three remarks here.

Introduction

Please include some information on the domestication history of tomato in your introduction and how the varieties you used relate to this. When did domestication begin, where? You mention the “degree of domestication”, suggesting that there is some kind of gradient between all three varieties but it does not become clear what you mean by this. In your previous response letter you mentioned that Micro-Tom has a smaller genome and presents genetic differences with other commercial lines. Please include some of this information in your manuscript and also mention what the genomic differences are.

Materials & methods

Concerning the bioassay with the psyllids: to me it is unclear whether the psyllids were removed after the 2-hour feeding period or whether they were left on the plants during the volatile collections. Please just include on sentence where this is clearly mentioned. Data analysis: multivariate ANOVA is the wrong term. I suppose you performed either one-factorial or two-factorial ANOVAs. In any case, these are univariate analyses. Please also state which non-parametric test/s was/were used.

Results

Whenever you mention the most abundant compounds, I think it is justified to mention the names instead of the numbers, especially since the lists will not be too long. I would also appreciate if you would give some kind of limit for what you call “were detected as abundant”. Is there a peak area cutoff which you would use (e.g. peak area > 108).

Tables/Figures.

General remarks 

Whenever you present the outcome of a statistical test in a table or in a figure, please also state in the respective figure caption or table heading which test has been used, e.g. Level of significance: *, P<0.05, t-test) In my opinion it would be good to mention the limit of detection of your GC-MC analysis, either also in each figure caption or, at least, in the methods section. I would prefer the first option though. Since all your graphs start at peak areas of 105, one may wonder if absence of a given bar, e.g. for compound 2 in Figure 2, means that the peak are was below 105 or whether the compound was not detected at all. Alternatively, you may also have your y-axes start at 100.

Specific remarks

Table 1:

Please state that log10-transformed peak areas are given in the table. Why is there no SD for some of the compounds mentioned in the table? Does that mean that they were only detected in one out of six replicate plants? In that case, I would also wonder how robust that specific response is. But at least, it should be mentioned that those compounds were only detected in one replicate. Table heading: in addition to the general remarks, please state that compounds that show statistically significant differences between healthy plants are shown in bold alongside with the respective test done and the level of significance used.

Table 2:

Please include the numbers that you assigned to each compound at the beginning of the first line. Please also include the t- or F- values you obtained together with the degrees of freedom. In the first line there is a dot missing for the P I still do not understand how you selected the compounds you show in your figures. In your previous response you say that they were selected for their abundance (>90%). Does that mean they were present in more than 90% of your replicates? Please clarify this and mention this in the figure caption, e.g. “Statistical analyses for the compounds that were shared in the headspace of healthy plants among at least two out of three varieties.” (if this was the case).

Figure 1:

Your figure caption states: “Fifteen compounds were detected and shared in the headspace of all varieties”. This is simply wrong, though. Please also state how the presented compounds were actually selected. Here, you could also include indications of the statistically significant differences you detected for compounds 18 and 20 as you did in the other two figures.

Figure 2:

Same as for figure 1

Figure 3:

Overall, I think this figure is a big improvement to the manuscript.

Please also state how the compounds that you present in the figure were selected. Are these the only ones you detected in a more than one infested plant? For compound 9 you present a value of 9.55 in healthy wild plants in table 1. However, in figure 3 compound 9 seems to only have been detected in infested wild plants. Please clarify! The same applies to compounds 11 and 22 for Floradade. If you decided to remove the compounds that were only detected in one replicate, maybe you might just filter your whole data set according to this criterion. This has been suggested earlier during the reviewing process, and according to your reply then, you have made these changes. But the fact that there are still values without SD in your table seem contradicting to this. For Micro-Tom you also mention L-beta-Pinene in the figure caption, although it is not shown in the figure. Pease remove.

Discussion

If you speculate that specific volatiles that you detected could act as attractants or repellents that might have mediated the oviposition preferences you have observed earlier, please mention which compounds these are, whether you would consider them attractive or repellent and whether there is any evidence from the scientific literature for these compounds serving as attractants or repellents. I admit that you do this, but this could be done in a much more structured way that would also make it easier for any reader to get the messag.

Author Response

Reviewer 1.

Comments on revised version of Mayo-Hernandez et al. – “Effects of Bactericera cockerelli herbivory on Voltaile Emissions of Three Varieties of Solanum lycopersicum (plants-580200_REV2_JHVS)

I admit that considerable improvements have been made. Language and grammar still need a revision, though. I will just mention a few examples:

l 36: “we found…”; please avoid subjective language.

R: Change done as suggested.

ll45: “…that domesticated tomato reduced the attraction or…”   imprecise language. Rather, the attraction of predators was reduced during the domestication process, or: domestication resulted in reduced predator attraction as compared to wild tomato varieties, etc…

R: Thanks for correction, changes were done as suggested.

l65: “markedly in the blends emitted by plants evaluated”

R: This mistake was corrected as indicated.

l197: “we found”

R: Done as requested.

ll227: “This report confirms that tomato varieties respond in different ways to psyllids infestation at different life stages.” This sentence implies that you examined whether tomato responses to psyllid infestation depend on psyllid life stage in the present study. But this is not the case.

R: We agree with this observation; we did not perform bioassays to evaluate others life stages of psyllid on tomato leaves. This sentence was removed in the manuscript.

Overall, Introduction and Material & methods seem fine to me. I only have three remarks here.

Introduction

Please include some information on the domestication history of tomato in your introduction and how the varieties you used relate to this. When did domestication begin, where? You mention the “degree of domestication”, suggesting that there is some kind of gradient between all three varieties but it does not become clear what you mean by this. In your previous response letter you mentioned that Micro-Tom has a smaller genome and presents genetic differences with other commercial lines. Please include some of this information in your manuscript and also mention what the genomic differences are.

R: The following paragraph was added to address this observation: Although the tomato was originated from Andean region in South America, the original site of domestication is undefined [10]. However, it is assumed that Mexico is the site of domestication and Peru as center of diversity [11]. Tomato (Solanum lycopersicum) is an economically important crop and is a model system for research. Among tomato models, a miniature dwarf cultivar Micro-Tom has been extensively studied for the last ten years, Micro-Tom has a small plant size, a short life cycle, a genome size (950 Mb) and represent an interesting genomic tool [12,13]. In other hand, wild tomato relative (Solanum pennelli) contain a larger genome size (1200 Mb) compared than commercial cultivars of tomato (approximately 950 Mb), this represent a dramatic genetic erosion in domesticated cultivars [14]. Therefore, the tomato domesticated are susceptible to phytopathogens and phytophagous insects, and resistance alleles are present in wild tomato relatives [15].”

Materials & methods

Concerning the bioassay with the psyllids: to me it is unclear whether the psyllids were removed after the 2-hour feeding period or whether they were left on the plants during the volatile collections. Please just include on sentence where this is clearly mentioned.

R: True. To clarify this point, the following line was inserted into the body of the MS: “(nymphs were left in plants)”.

Data analysis: multivariate ANOVA is the wrong term. I suppose you performed either one-factorial or two-factorial ANOVAs. In any case, these are univariate analyses. Please also state which non-parametric test/s was/were used.

R: We regret this mistake. To correct it, the text in Data analysis was changed.

Results

 Whenever you mention the most abundant compounds, I think it is justified to mention the names instead of the numbers, especially since the lists will not be too long. I would also appreciate if you would give some kind of limit for what you call “were detected as abundant”. Is there a peak area cutoff which you would use (e.g. peak area > 108).

R: We agree with this observation. Changes were included as requested.

Tables/Figures.

General remarks 

Whenever you present the outcome of a statistical test in a table or in a figure, please also state in the respective figure caption or table heading which test has been used, e.g. Level of significance: *, P<0.05, t-test)

R: Changes made as suggested.

In my opinion it would be good to mention the limit of detection of your GC-MC analysis, either also in each figure caption or, at least, in the methods section. I would prefer the first option though. Since all your graphs start at peak areas of 105, one may wonder if absence of a given bar, e.g. for compound 2 in Figure 2, means that the peak are was below 105 or whether the compound was not detected at all. Alternatively, you may also have your y-axes start at 100.

R: Done as requested.

Specific remarks

Table 1:

Please state that log10-transformed peak areas are given in the table.

R: Changes done as requested.

Why is there no SD for some of the compounds mentioned in the table? Does that mean that they were only detected in one out of six replicate plants? In that case, I would also wonder how robust that specific response is. But at least, it should be mentioned that those compounds were only detected in one replicate.

R: Change done as requested: “Data with no SD were only detected in one replicate”.

Table heading: in addition to the general remarks, please state that compounds that show statistically significant differences between healthy plants are shown in bold alongside with the respective test done and the level of significance used.

R: Change introduced as suggested.

Table 2:

Please include the numbers that you assigned to each compound at the beginning of the first line.

R: Changes done as requested.

Please also include the t- or F- values you obtained together with the degrees of freedom.

R: Done as requested.

In the first line there is a dot missing for the P

R: We acknowledge the error. Corrected as suggested.

I still do not understand how you selected the compounds you show in your figures. In your previous response you say that they were selected for their abundance (>90%). Does that mean they were present in more than 90% of your replicates? Please clarify this and mention this in the figure caption, e.g. “Statistical analyses for the compounds that were shared in the headspace of healthy plants among at least two out of three varieties.” (if this was the case).

R: Changes made as requested. To clarify this point, the following lines were inserted in the manuscript: Statistical analyses for the compounds that were shared in the headspace of healthy plants among at least two out of three varieties (those compounds with > 90% of quality were selected).”

Figure 1:

Your figure caption states: “Fifteen compounds were detected and shared in the headspace of all varieties”. This is simply wrong, though. Please also state how the presented compounds were actually selected.

R: Change done as suggested.

Here, you could also include indications of the statistically significant differences you detected for compounds 18 and 20 as you did in the other two figures.

R: We agree with this observation but no differences were detected in both compounds.

Figure 2:

Same as for figure 1

R: Changes done as requested.

Figure 3:

Overall, I think this figure is a big improvement to the manuscript.

Please also state how the compounds that you present in the figure were selected. Are these the only ones you detected in a more than one infested plant?

R: We agree with this observation. Those compounds with replicates (at least n=3) were selected and were compared in healthy vs infested plants.

For compound 9 you present a value of 9.55 in healthy wild plants in table 1. However, in figure 3 compound 9 seems to only have been detected in infested wild plants. Please clarify! The same applies to compounds 11 and 22 for Floradade. If you decided to remove the compounds that were only detected in one replicate, maybe you might just filter your whole data set according to this criterion. This has been suggested earlier during the reviewing process, and according to your reply then, you have made these changes. But the fact that there are still values without SD in your table seem contradicting to this.

R: We agree with this observation. To clarify this point, no replicates were detected for compounds 5, 9, 11, and 22 in healthy plants. Although in table 1 was introduced the following line: “Data with no SD were only detected in one replicate”.

For Micro-Tom you also mention L-beta-Pinene in the figure caption, although it is not shown in the figure. Pease remove.

R: We regret this mistake. To correct it, the figure caption was changed.

Discussion

If you speculate that specific volatiles that you detected could act as attractants or repellents that might have mediated the oviposition preferences you have observed earlier, please mention which compounds these are, whether you would consider them attractive or repellent and whether there is any evidence from the scientific literature for these compounds serving as attractants or repellents. I admit that you do this, but this could be done in a much more structured way that would also make it easier for any reader to get the messag.

R: In L213-217, we mentioned the possible repellence of two compounds in wild plants: “VOCs induced in wild plants, β-phellandrene and (+)-4-carene were detected as the most abundant (Table 1), this could correlate with the reduced oviposition of B. cockerelli [25] and also with the role of monoterpenes involved with repellence in solanaceous plants against whiteflies, such as terpinenes and phellandrenes [9]”.

Reviewer 2 Report

The authors were able to address some of my concerns in the revision.  However, they generated some additional questions. 

What was the "nonparametric" analysis that was conducted and which data were evaluated using the nonparametric test(s)?  It was not clear to me how statistical tests were conducted with only one "replicate", unless each plant was considered a separate replicate in some tests while in other tests they used groups of two or three plants as a replicate.  I suspect some of this may be a language problem. Regardless, had the tests used a balanced experimental design, the methodology, statistical analyses, and conclusions would have been much easier to understand.

Author Response

Reviewer 2.

Comments and Suggestions for Authors

The authors were able to address some of my concerns in the revision.  However, they generated some additional questions. 

What was the "nonparametric" analysis that was conducted and which data were evaluated using the nonparametric test(s)?  It was not clear to me how statistical tests were conducted with only one "replicate", unless each plant was considered a separate replicate in some tests while in other tests they used groups of two or three plants as a replicate.  I suspect some of this may be a language problem. Regardless, had the tests used a balanced experimental design, the methodology, statistical analyses, and conclusions would have been much easier to understand.

R: We regret this mistake. To correct it, the manuscript was modified according the suggested.
